EMBO
Molecular Medicine

# CoA-dependent activation of mitochondrial acyl carrier protein links four neurodegenerative diseases

Roald A Lambrechts[1], Hein Schepers[1,†], Yi Yu[1,†], Marianne van der Zwaag[1], Kaija J Autio[2], Marcel A Vieira-Lara[3], Barbara M Bakker[3], Marina A Tijssen[4], Susan J Hayflick[5] (iD), Nicola A Grzeschik[1] & Ody CM Sibon[1,*] (iD)

## Abstract

PKAN, CoPAN, MePAN, and PDH-E2 deficiency share key phenotypic features but harbor defects in distinct metabolic processes. Selective damage to the globus pallidus occurs in these genetic neurodegenerative diseases, which arise from defects in CoA biosynthesis (PKAN, CoPAN), protein lipoylation (MePAN), and pyruvate dehydrogenase activity (PDH-E2 deficiency). Overlap of their clinical features suggests a common molecular etiology, the identification of which is required to understand their pathophysiology and design treatment strategies. We provide evidence that CoA-dependent activation of mitochondrial acyl carrier protein (mtACP) is a possible process linking these diseases through its effect on PDH activity. CoA is the source for the 4′-phosphopantetheine moiety required for the posttranslational 4′-phosphopantetheiny-lation needed to activate specific proteins. We show that impaired CoA homeostasis leads to decreased 4′-phosphopantetheinylation of mtACP. This results in a decrease of the active form of mtACP, and in turn a decrease in lipoylation with reduced activity of lipoylated proteins, including PDH. Defects in the steps of a linked CoA-mtACP-PDH pathway cause similar phenotypic abnormalities. By chemically and genetically re-activating PDH, these phenotypes can be rescued, suggesting possible treatment strategies for these diseases.

**Keywords** 4′-phosphopantetheinylation; Coenzyme A; mtACP; NBIA; NDUFAB1

**Subject Categories** Pharmacology & Drug Discovery; Neuroscience

See also: **SY Jeong et al** (December 2019)

## Introduction

Coenzyme A (CoA) is an essential cofactor participating in approximately 9% of all cellular metabolic reactions, such as the tricarboxylic acid (TCA) cycle, and fatty acid synthesis and degradation (Leonardi et al, 2005; Strauss, 2010). CoA is synthesized de novo in cells, utilizing vitamin B5 as a starting molecule and requiring five enzymatic reactions. These are carried out by pantothenate kinase (PANK), phosphopantothenoylcysteine synthetase (PPCS), phospho-pantothenoylcysteine decarboxylase (PPCDC), phosphopantetheine adenylyltransferase (PPAT), and dephospho-CoA kinase (DPCK), respectively (Leonardi et al, 2005; Strauss, 2010). In some organisms, including Drosophila melanogaster, mice, and humans, PPAT and DPCK enzyme activities are performed by a single bifunctional protein, referred to as CoA synthase or COASY (Fig 1). The intermediate products that are being sequentially formed from vitamin B5 during the CoA de novo biosynthesis pathway are as follows: 4′-phosphopantothenate, 4′-phosphopantothenoylcysteine, 4′-phospho-pantetheine, dephospho-CoA, and CoA (Fig 1). Enzymes of the CoA de novo biosynthesis pathway are evolutionarily conserved, further underscoring the importance of this pathway for all living organisms.

Two autosomal recessive neurodegenerative diseases are caused by mutations in genes encoding enzymes of the CoA pathway. Pathogenic variants in PANK2 and COASY lead to two early-onset neurodegenerative diseases: pantothenate kinase-associated neurodegeneration (PKAN) and CoA synthase protein-associated neurodegeneration (CoPAN). The human genome contains four genes encoding pantothenate kinase homologs, PANK1-4, and only mutations in PANK2 are associated with PKAN. PKAN and CoPAN patients accumulate iron in the globus pallidus, a basal ganglia structure in the brain (Hayflick et al, 2003; Dusi et al, 2014). Iron accumulation is visible on T2-weighted imaging as a hypointense signal on MRI. In PKAN and CoPAN areas, T2-hyper-intense signals are also seen at the globus pallidus, indicating edema and tissue damage (Kruer et al, 2012). These CoA-linked

1 Department of Biomedical Sciences of Cells and Systems, University Medical Center Groningen, University of Groningen, Groningen, The Netherlands
2 Faculty of Biochemistry and Molecular Medicine, University of Oulu, Oulu, Finland
3 Laboratory of Pediatrics, Section Systems Medicine of Metabolism and Signaling, University Medical Center Groningen, University of Groningen, Groningen, The Netherlands
4 Neurology Department, University Medical Center Groningen, University of Groningen, Groningen, The Netherlands
5 Molecular and Medical Genetics, Oregon Health & Science University, Portland, OR, USA
  *Corresponding author. Tel: +31 503616111; E-mail: o.c.m.sibon@umcg.nl
  †These authors contributed equally to this work

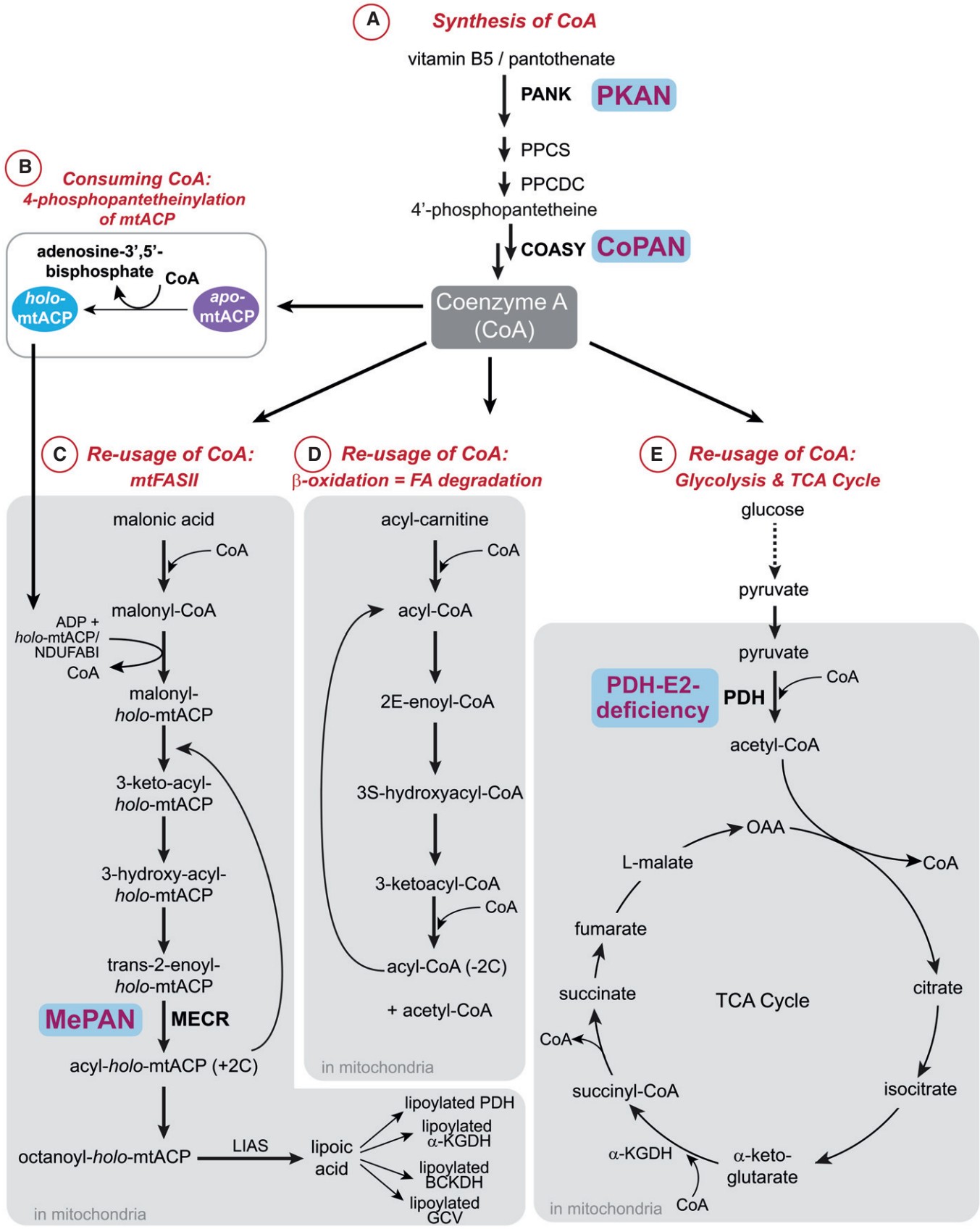

**Figure 1.**

◀

**Figure 1. Metabolic pathways in which coenzyme A is formed, re-used, or consumed and their interconnections.**

A *De novo* biosynthesis pathway of coenzyme A (CoA) is a pathway during which CoA is produced. Vitamin B5 is taken up by cells and converted into CoA by the action of five enzymatic reactions (Leonardi *et al*, 2005; Strauss, 2010). These are carried out by pantothenate kinase (PANK), phosphopantothenoylcysteine synthetase (PPCS), phosphopantothenoylcysteine decarboxylase (PPCDC), phosphopantetheine adenylyltransferase (PPAT), and dephospho-CoA kinase (DPCK), respectively. In *Drosophila melanogaster*, mice, and humans, PPAT and DPCK enzyme activities are carried out by a single bifunctional protein, CoA synthase or COASY. Abbreviations of the enzymes are provided. The starting product vitamin B5/pantothenate, the intermediate, 4′-phosphopantetheine, and the final product CoA are depicted. PKAN and CoPAN are inherited recessive diseases caused by homozygous mutations in PANK2 and COASY, respectively.

B Formation of *holo*-mtACP (active form of mitochondrial acyl carrier protein) is a CoA consuming metabolic reaction (Beld *et al*, 2014). 4′-phosphopantetheinylation of inactive *apo*-mtACP is required to produce the active *holo*-mtACP form. During this process, CoA serves as the source for the 4′-phosphopantetheinylation, and hereby, a CoA molecule is degraded and adenosine 3′–5′-biphosphate is released. *Holo*-mtACP plays a key role in mitochondrial fatty acid synthesis, which can be visualized in (C).

C Fatty acid synthesis is a metabolic pathway in which CoA is re-used (Kastaniotis *et al*, 2017). Mitochondrial fatty acid synthesis (mtFASII) is required for the synthesis of lipoic acid, a pathway that is dependent on the activity of *holo*-mtACP. In this pathway, malonic acid is converted to malonyl-CoA, which requires *holo*-mtACP (or NDUFABI in humans) for the formation of malonyl-*holo*-ACP as well as for subsequent downstream steps. Mitochondrial Enoyl-[acyl-carrier-protein] reductase (MECR, defective enzyme in MEPAN) is required for the formation of acyl-*holo*-mtACP as indicated. Via this pathway lipoic acid is formed from octanoyl-*holo*-mtACP by lipoic acid synthetase (LIAS). This product is then required for the formation of lipoylated pyruvate dehydrogenase (PDH), lipoylated α-ketoglutarate dehydrogenase (αKGDH), lipoylated branched-chain alpha-keto acid dehydrogenase (BCKDH), and lipoylated glycine cleavage system (GCV). Lipoylation of the above-mentioned proteins is necessary for catalysis of their respective reactions to occur.

D Fatty acid (FA) degradation or β-oxidation is an example of a degradation pathway in which CoA is re-used. The starting mitochondrial precursor acyl-carnitine, the intermediates, and the end product of this pathway are indicated, as well as reactions in which CoA is required (but not consumed) and released.

E In the glycolysis and TCA cycle, CoA is re-used. Pyruvate dehydrogenase (PDH) catalyzes the oxidative decarboxylation of pyruvate to produce acetyl that is coupled to CoA to produce acetyl-CoA. Impaired function of this enzyme leads to PDH-E2 deficiency. The product of the PDH reaction, acetyl-CoA, is the fuel for the TCA cycle which is also a CoA re-using pathway as indicated. OAA: oxaloacetate.

diseases are characterized by progressive motor dysfunction and severe dystonia.

Damage to the globus pallidus occurs in other inborn errors of metabolism as well. MePAN, a third childhood-onset neurodegenerative disorder, manifests with damage to the globus pallidus, which is also visible on brain MRI as hyperintense signal on T2-weighted imaging although without the iron-associated signal abnormalities (Heimer *et al*, 2016). MePAN patients carry mutations in the gene encoding mitochondrial enoyl-[acyl-carrier-protein] reductase (MECR), one of four enzymes involved in the elongation of fatty acids in mitochondria to form octanoic acid, a precursor of lipoic acid (Fig 1; Heimer *et al*, 2016). In eukaryotic cells, this process of mitochondrial fatty acid synthesis (mtFAS-type II) is required for lipoic acid production and subsequently for lipoylation of proteins (Brody *et al*, 1997; Wada *et al*, 1997; Feng *et al*, 2009). Finally, mutations causing impairment of a component of the pyruvate dehydrogenase complex, PDH-E2, lead to PDH-E2 deficiency and cause a form of Leigh disease, in which neuroradiographic abnormalities are again observed specifically in the globus pallidus (Head *et al*, 2005; McWilliam *et al*, 2010; Leoni *et al*, 2012). PDH catalyzes the oxidative decarboxylation of pyruvate to produce acetyl-CoA, thereby linking glycolysis to the TCA cycle (Fig 1). The symptoms, signs, and MRI characteristics of PKAN and PDH-E2 deficiency can be similar. In patients with a clinical suspicion of PKAN, PDH-E2 deficiency should also be considered in the differential diagnosis and *vice versa* (Head *et al*, 2005; McWilliam *et al*, 2010; Leoni *et al*, 2012). The clinical and neuroradiographic overlapping features of PKAN, CoPAN, MePAN, and PDH-E2 deficiency suggest a common element in their pathogeneses.

The pathways of CoA biosynthesis, mitochondrial fatty acid synthesis, and glycolysis/TCA cycle show interdependency and interconnectivity (Leonardi *et al*, 2005; Strauss, 2010; Beld *et al*, 2014; Kastaniotis *et al*, 2017; Fig 1), but a specific molecular etiology common to these four disorders has been lacking.

Herein, we propose a common underlying pathway directly connecting these four diseases. In most CoA-dependent metabolic reactions, CoA acts as an acyl carrier; the acyl moiety is transferred between CoA and another molecule (e.g., carnitine), leaving CoA intact and available for further transfer reactions. CoA-dependent acyl transfer occurs during the TCA cycle, fatty acid synthesis, and fatty acid degradation (Fig 1). Because CoA is re-used as acyl carrier component, these reactions do not lead to reduced levels of total CoA. In contrast, one specific form of posttranslational modification "consumes" CoA. 4′-phosphopantetheinylation adds a 4′-phospho-pantetheine moiety to specific proteins resulting in their activation. For this modification, the 4′-phosphopantetheine moiety is derived from CoA (Beld *et al*, 2014). Therefore, this reaction forms a 4′-phosphopantetheinylated protein, and adenosine 3′,5′-bisphosphate is released and thereby causes a net loss of CoA (Elovson & Vagelos, 1968). In humans, a select group of proteins requires this 4′-phosphopantetheine moiety in order to function, e.g. 10-formyltetrahydrofolate dehydrogenase, an enzyme of folate metabolism (Strickland *et al*, 2010), cytosolic fatty acid synthase, and mitochondrial acyl carrier protein (mtACP; Joshi *et al*, 2003; Beld *et al*, 2014). We focussed on mtACP because human PANK2, COASY, MECR, and PDH-E2 are mitochondrial proteins (Kotzbauer *et al*, 2005; Dusi *et al*, 2014) and mitochondria are defective in various PKAN animal models (Rana *et al*, 2010; Brunetti *et al*, 2014; Orellana *et al*, 2016; Jeong *et al*, 2019). The active 4′-phosphopantetheinylated form of mtACP is referred to as *holo*-mtACP. mtACP, known in humans as NDUFAB1, is one of the subunits of the respiratory chain complex I and plays a central role in mitochondrial fatty acid synthesis (Brody *et al*, 1997; Feng *et al*, 2009). In this latter process, the thiol group of the 4′-phosphopantetheine prosthetic group forms the attachment site for a growing carbon chain (Beld *et al*, 2014). Octanoate formed in this way is converted to lipoic acid and used to modify mitochondrial proteins, among which are the E2 subunits of three enzyme complexes (pyruvate dehydrogenase (PDH), α-ketoglutarate dehydrogenase (αKGDH), branched-chain α-keto acid dehydrogenase (BCKDH), and the glycine cleavage system (GCV); Cronan, 2016; Rowland *et al*, 2018). In humans, octanoic acid is transferred to its target proteins by the lipoyl transferases

LIPT2 and LIPT1. By the action of the conserved enzyme lipoic acid synthase (*LIAS* in humans), protein-bound octanoate is transformed into lipoic acid (LA) by the insertion of two sulfhydryl groups (Booker, 2004; Hiltunen *et al*, 2010; Solmonson & DeBerardinis, 2018), enabling the now-lipoylated proteins to function (Fig 1;

Hiltunen *et al*, 2010). Based on these reports, we hypothesized that, because 4′-phosphopantetheinylation of mtACP consumes CoA, this reaction may be most sensitive to impaired CoA biosynthesis. Consequently, downstream processes are predicted to be affected as well, including decreased lipoylation and activity of PDH (Fig 2).

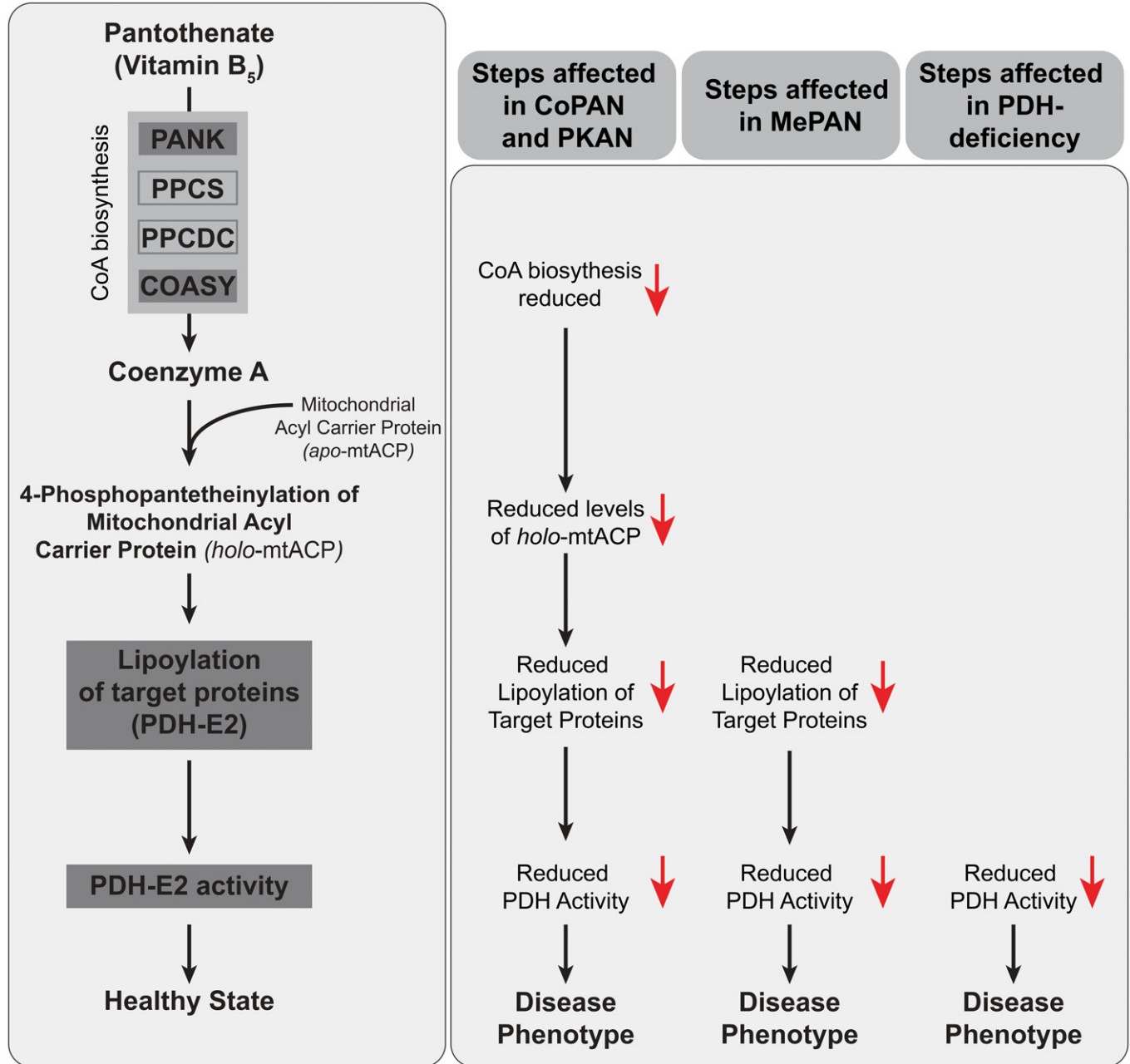

**Figure 2. *De novo* CoA biosynthesis pathway and key downstream steps to link PKAN, CoPAN, MePAN, and PDH-E2 deficiency.**

Left part: Proposed linear pathway linking CoA-mtACP-PDH. From top to bottom: The *de novo* CoA biosynthesis pathway starts with the cellular uptake of pantothenate (Vitamin B5). Pantothenate kinase (PANK), phosphopantothenoylcysteine synthetase (PPCS), phosphopantothenoylcysteine decarboxylase (PPCDC), and coenzyme A synthase (COASY) are enzymes required for the *de novo* biosynthesis of CoA. Mitochondrial acyl carrier protein (mtACP) undergoes a posttranslational modification and active *holo*-mtACP is formed. This posttranslational modification consists of 4′-phosphopantetheine, which is derived from CoA. *Holo*-mtACP in turn is required for lipoylation of PDH-E2, a modification necessary for activation of the PDH complex. It is hypothesized that a decrease in CoA biosynthesis leads to decreased amounts of *holo*-ACP, decreased lipoylation of PDH-E2, and decreased activity of PDH. Right part: Steps that are affected in PKAN, CoPAN, MePAN, and PDH-E2 deficiency. Primary affected steps caused by the genetic defects are depicted as the most upstream steps as well as the hypothesized downstream steps for CoPAN, PKAN, and MePAN.

This hypothesis could explain how defects in PANK2 and COASY might lead to a phenotype similar to that arising from defects in MECR and PDH (Fig 2). A central tenet of this hypothesis is that impaired *de novo* biosynthesis of CoA leads to decreased levels of *holo*-mtACP. Here, we investigated this hypothesis.

The *de novo* CoA biosynthesis pathway and the metabolic reactions presented in Figs 1 and 2) are highly conserved between human and *Drosophila melanogaster* (Leonardi *et al*, 2005; Bosveld *et al*, 2008). Our combined approach employing the versatile genetic tools of *Drosophila melanogaster* and mammalian cells enabled us to demonstrate that impaired CoA biosynthesis leads to decreased levels of active, 4′-phosphopantetheinylated mtACP. This observation was associated with decreased lipoylation of PDH-E2 and decreased PDH activity. Our results revealed the presence of a CoA-mtACP-PDH pathway in which the 4′-phosphopantetheinylation of mtACP is a key step. Next, we showed that stimulation of PDH rescued phenotypes caused by impaired CoA biosynthesis, highlighting PDH as a possible common target for ameliorating diseases induced by defects in the CoA-mtACP-PDH pathway. Our findings combined with those reported by Jeong *et al* suggest therapeutic approaches for PKAN, CoPAN, MePAN, and PDH-E2 deficiency.

# Results

### *holo*-mtACP levels are reduced by impeding CoA biosynthesis

PKAN and CoPAN patients carry mutations in genes coding for pantothenate kinase 2 and COASY, enzymes required for CoA biosynthesis (Fig 1). To test our hypothesis (presented in Fig 2), we first investigated consequences of impaired CoA biosynthesis on 4′-phosphopantetheinylation of mtACP. For this, we chose *Drosophila melanogaster* because of its conserved metabolic steps and genes and its versatile genetic tools. mtACP requires activation in order to function; the active *holo* form is generated by enzymatic transfer of a negatively charged 4′-phosphopantetheine moiety to a conserved serine residue of the inactive *apo* form (Elovson & Vagelos, 1968; Jung *et al*, 2016; Fig 3A). For our study we manipulated *mtacp*, the *Drosophila melanogaster* gene encoding mtACP, containing Ser-99, which is predicted to bind 4-phosphopantetheine (Ragone *et al*, 1999). In order to be able to identify and distinguish the two forms of mtACP (*holo* versus *apo*), we generated constructs encoding mutant proteins that would be refractory to 4-phosphopantetheinylation and observed their mobility differences using gel electrophoresis. We mutated the crucial serine residue: one to mimic the uncharged *apo* form (S99A) and two negatively charged forms (S99D and S99E) to mimic the charged *holo* form of mtACP (Fig 3A). Overexpression of wild-type mtACP constructs in *Drosophila* S2 Schneider cells enabled the visualization of protein bands that correspond to endogenous *apo*- and *holo*-mtACP forms. By comparing these bands to the *apo*-mimetic S99A and *holo*-mimetics S99D and S99E, we were able to prove the identity of the bands visualized under control and mtACP wild-type overexpressing conditions. Under physiologic conditions, endogenous *holo*-mtACP was detected (Fig 3B and C). In contrast, no endogenous *apo*-mtACP protein was visible, consistent with previous observations in other organisms that the inactive *apo*-mtACP form is not stable (Jackowski & Rock, 1983; Post-Beittenmiller *et al*, 1989).

We proceeded to investigate whether levels of active *holo*-mtACP would decrease upon CoA deprivation, a key assumption of our hypothesis. Treating S2 cells with the PANK inhibitor hopantenate (HoPan) leads to reduced CoA biosynthesis and to reduced levels of total CoA (Rana *et al*, 2010; Siudeja *et al*, 2011; Srinivasan *et al*, 2015). Under these conditions, we observed reduced levels of endogenous *holo*-mtACP (Fig 3C), and addition of CoA to the medium of HoPan-treated cells reverted this phenotype, consistent with previous studies in which administration of extracellular CoA is able to rescue phenotypes associated with reduced intracellular CoA biosynthesis (Rana *et al*, 2010; Siudeja *et al*, 2011; Srinivasan *et al*, 2015). These results demonstrate that impaired CoA biosynthesis is associated with decreased levels of 4′-phosphopantetheinylation of mtACP.

### Protein lipoylation is reduced by impeding CoA biosynthesis

To further investigate the consequences of impaired CoA biosynthesis and reduced levels of 4′-phosphopantetheinylated mtACP, we examined mtACP-dependent processes, specifically those linked to MePAN and PDH-E2 deficiency. *holo*-mtACP and MECR are required for protein lipoylation (Fig 1), a process that is therefore affected in MePAN patients (Heimer *et al*, 2016). Four evolutionary conserved lipoylated enzyme complexes have been identified: PDH, α-KGDH, BCKDH, and GCV. To assess whether lipoylation was affected under conditions of impaired CoA biosynthesis, total protein lipoylation was analyzed using Western blot analysis. Incubation with an antibody that detects protein-bound lipoic acid revealed decreased levels of various protein bands under conditions of reduced CoA levels, an effect that was rescued when CoA was supplemented to the medium (Fig 3D, Appendix Fig S1). The availability of an antibody recognizing the *Drosophila* PDH-E2 subunit allowed the analysis of lipoylated PDH-E2 specifically, demonstrating that lipoylation of PDH-E2 was compromised in a CoA reduced background compared to control levels, an effect that was rescued by replenishing CoA (Fig 3D and E). Specific antibodies for the other lipoylated fly enzymes are lacking, but the observation that total protein lipoylation is reduced under conditions of impaired CoA biosynthesis suggests that the other lipoylated fly enzymes are similarly affected.

### Pyruvate dehydrogenase activity decreases upon CoA deprivation

Because we were able to specifically document decreased PDH lipoylation under reduced CoA biosynthesis conditions, we predicted that PDH enzyme activity would be decreased. Lipoylation of PDH-E2 is essential for the subunit and complex to function, as the lipoyl moiety binds and oxidizes the hydroxyl-ethyl compound derived from pyruvate and subsequently transfers it to its acceptor CoA to generate acetyl-CoA (Patel *et al*, 2014). To determine whether CoA deprivation indeed decreases PDH activity, we quantified PDH activity in control S2 cells and cells treated with HoPan (Fig 3F). We observed a significant reduction in PDH activity upon HoPan treatment compared to control cells, an effect rescued by addition of CoA to the HoPan-treated cells (Fig 3F). We assumed that, in the HoPan-treated cells, a fraction of the remaining pool of lipoylated PDH is partly inactivated by the presence of the endogenous PDH-inhibitor pyruvate dehydrogenase kinase (PDK). Therefore, we predicted that residual PDH activity could be increased by

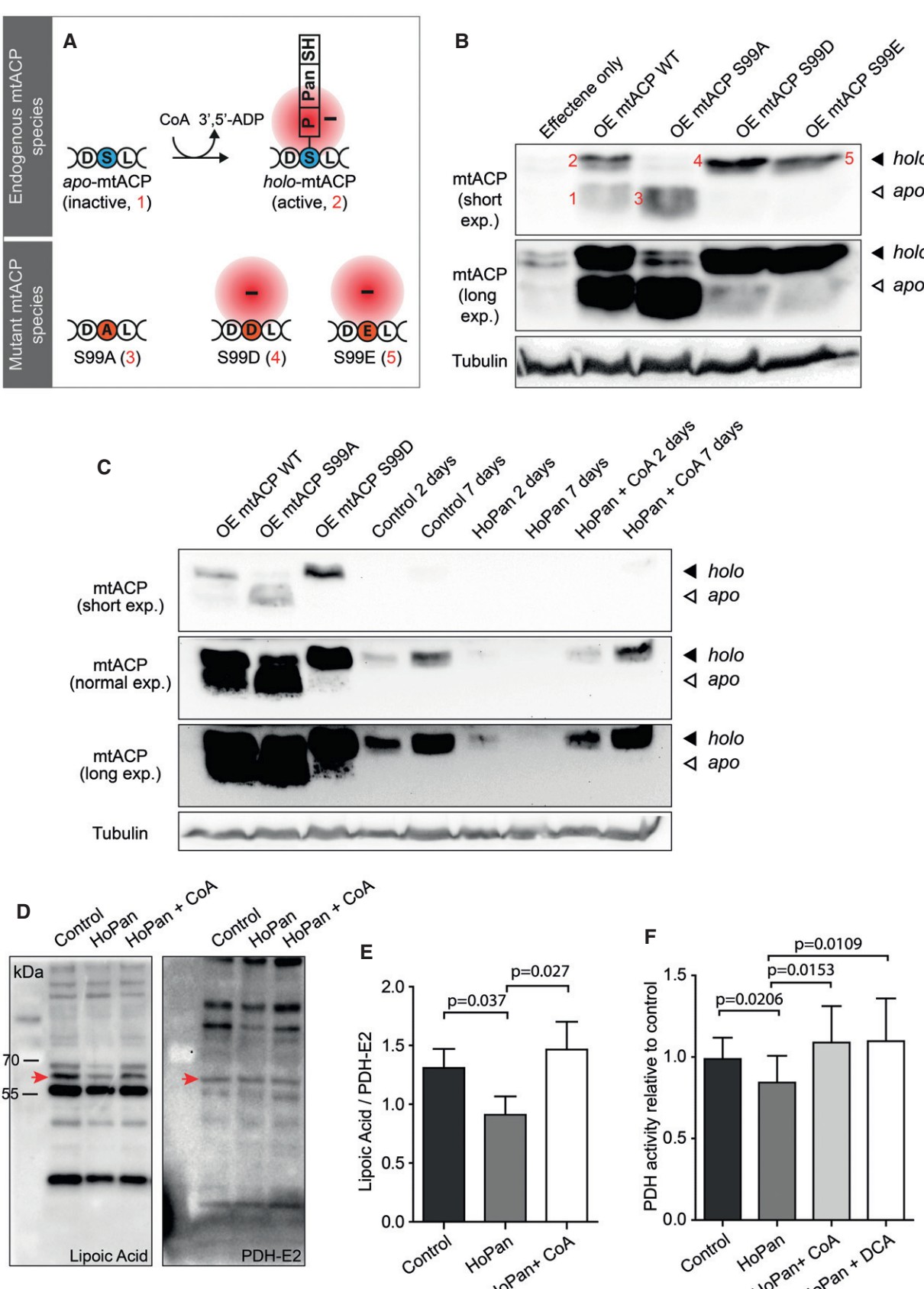

Figure 3.

**Figure 3. Decreased levels of CoA are associated with decreased levels of *holo*-mtACP and lipoylated PDH-E2.**

A   Schematic presentation of endogenous active and inactive forms of mtACP and synthesized mutant forms of mtACP. For the endogenous form, the S represents the serine residue that is 4′-phosphopantetheinylated, while D and L represent the flanking amino acids. Inactive form of mtACP (*apo*-mtACP) is indicated with 1. CoA is the source for 4′-phosphopantetheine and is required for 4′-phosphopantetheinylation of mtACP occurring on the serine residue. This posttranslational modification results in an active form of mtACP, *holo*-mtACP, which is negatively charged and indicated with 2. Three mtACP constructs were generated: One in which serine 99 was modified to an alanine, indicated with a 3 and indicated as S99A (non-4′-phosphopantetheinylatable form); one in which serine 99 was modified into aspartate, indicated with a 4 and indicated as S99D (phosphomimetic); one in which serine 99 was modified into glutamate, indicated with a 5 and indicated as S99E (phosphomimetic). S99D and S99E are negatively charged, mimicking the negatively charged *holo*-mtACP. The red circle indicates the presence of a negative charge.

B   Western blot analysis of S2 cells overexpressing wild-type constructs of mtACP or the various mutant forms. First lane: lysates of control cells, resulting in the detection of *holo*-mtACP visible after long exposure. Second lane: overexpression (OE) of mtACP WT results in the detection of an *apo*-mtACP form and a *holo*-mtACP form, indicated with 1 and 2, respectively. Third lane: overexpression of mtACP S99A mutant form resulted in the detection of an *apo*-mtACP (non-4′-phosphopantetheinylatable) band only, indicated with 3. Fourth lane: overexpression of mtACP S99D resulted in the detection of a phosphomimetic form of mtACP only, migrating at the same mobility as *holo*-mtACP, indicated with 4. Fifth lane: overexpression of mtACP S99E results in the detection of a phosphomimetic form of mtACP only, migrating at the same mobility as *holo*-mtACP, indicated with 5. For visualization, a low exposure and high exposure blot are shown. Note that overexposed blots were required to visualize endogenous forms of mtACP.

C   Western blot analysis of mtACP forms under control conditions, under conditions of HoPan treatment, and under conditions of HoPan + CoA treatment. Lanes showing overexpression of mtACP WT, mtACP S99A, and mtACP S99D were used to allow identification of the *holo*- and the *apo-forms* of mtACP. α-Tubulin was used as a loading control. Various exposure times of the blots are presented to allow identification of mtACP under all conditions.

D   Western blot analysis showing lipoylated proteins under control conditions, after HoPan treatment and after HoPan + CoA treatment. S2 cells were treated with HoPan or HoPan + CoA for 4 days; non-treated cells were used as control. Antibodies specifically recognizing lipoylated proteins or PDH-E2 were used. Arrow heads indicate lipoylated PDH-E2 (left panel) or total PDH-E2 (right panel).

E   Quantification of lipoylated PDH-E2 from (D). Mean ± SD is given. *n* = 3 for all samples.

F   PDH activity was measured in control cells, HoPan-treated cells or HoPan-treated cells rescued with CoA or DCA. Mean ± SD of five biological replicates, each composed of three technical replicates and corrected for protein concentration.

Data information: For (E and F), two–tailed Student's *t*-test was performed to calculate statistical significance for the indicated subsets.
Source data are available online for this figure.

interfering with the action of this inhibitory enzyme. To test this, we added dichloroacetate (DCA) to the medium of the HoPan-treated cells. DCA is a clinically used drug (Stacpoole *et al*, 1983, 1992) that inhibits the PDH inhibitor, PDK (Walsh *et al*, 1976; Patel & Korotchkina, 2006), and as such leads to activation of PDH (Whitehouse *et al*, 1974). Addition of DCA indeed caused an increase in PDH activity compared to HoPan-treated cells (Fig 3F). These results indicate that the pyruvate dehydrogenase complex is functionally impaired as a consequence of CoA biosynthesis inhibition and under these circumstances can still be activated by DCA. These data support the existence of a CoA-mtACP-PDH pathway (Fig 2).

**Direct stimulation of PDH rescues phenotypes induced by *dPANK/fbl* knockdown *in vivo***

Our next step was to investigate a causal connection between CoA dyshomeostasis and PDH activity in a multicellular organism, *Drosophila melanogaster*, by asking whether phenotypes induced by impaired CoA biosynthesis could be rescued by recovery of PDH activity. To test this in whole organisms and complement our studies in *Drosophila* S2 cells, we used a genetic approach in flies, employing the binary *UAS/GAL4* system (Brand & Perrimon, 1993; Dietzl *et al*, 2007) to knockdown *dPANK/fbl* using RNAi. *dPANK/fbl* is the *Drosophila* ortholog of human *PANK2* (Afshar *et al*, 2001; Bosveld *et al*, 2008). In contrast to mammals, *Drosophila* possesses only one *pank* gene (Afshar *et al*, 2001). However, this gene gives rise to several *pank* isoforms, among which is one that localizes to mitochondria (Wu *et al*, 2009). Ubiquitous expression of a *dPANK/fbl-RNAi* construct in all tissues of the fly using an actin promotor resulted in significantly decreased *dPANK/fbl* mRNA and protein in two independent lines (Fig 4A, Appendix Fig S2), demonstrating the effectiveness of the *RNAi* constructs. Downregulation of *dPANK/fbl*

resulted in fewer flies that reached the adult stage compared to control flies (Fig 4B, Appendix Fig S3). Impairment of *dPANK/fbl* in *Drosophila* results in decreased total CoA levels and reduced viability (Rana *et al*, 2010; Srinivasan *et al*, 2015). Addition of pantethine to the fly food, which restores total CoA levels in a *dPANK/fbl*-deprived background (Rana *et al*, 2010), fully restored their viability, further confirming the CoA dependency of this phenotype (Fig 4B). Next we investigated whether stimulation of PDH in the *dPANK/fbl* knockdown flies increased their viability as well. For this, we added DCA to the fly food and consistent with the results obtained from S2 cells, a dose-dependent restoration of viability was observed (Fig 4B). These results suggest that viability can be recovered by boosting the activity of PDH.

**Downregulation of key steps of the CoA-PDH pathway during development causes a common abnormal morphogenetic phenotype in *Drosophila* wings**

The dose-dependent rescue by DCA of the *dPANK/fbl* knockdown phenotype provides evidence that deleterious phenotypes associated with CoA deprivation are at least partially mediated via decreased activity of PDH. The proposed pathway (Fig 2) further predicts that impairment of individual components of this pathway would cause a phenotype with overlapping characteristics. To investigate this, we created a genetically and phenotypically tractable system. Using tissue-specific RNAi-mediated knockdown of various genes along the CoA-mtACP-PDH pathway, we selected the wing as our target tissue. Visible details of wing structure and morphogenesis are well-delineated and provide a robust and quantifiable phenotypic readout. Importantly, disruption of metabolism in specific cells of the developing wing does not cause lethality of the organism, as is common when critical metabolic pathways are disrupted in other organ systems or in whole organisms. During early *Drosophila*

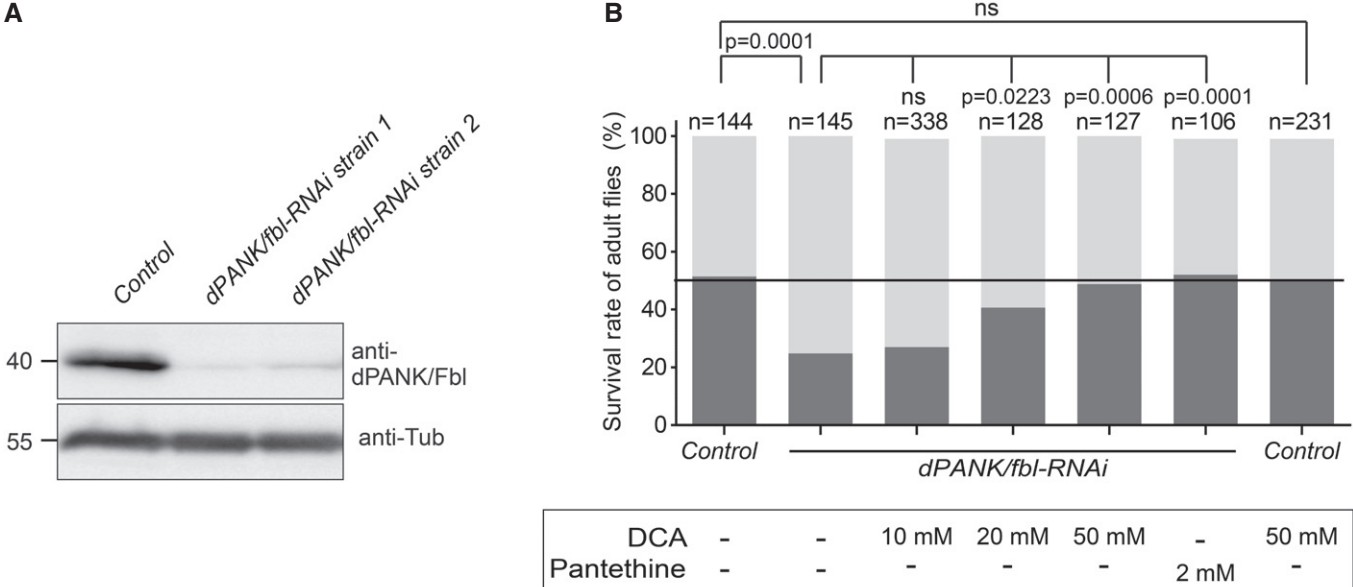

**Figure 4.** Downregulation of *dPANK/fbl* induces decreased viability of adult flies, which is rescued by pantethine and DCA treatment.

A Decreased dPANK/Fbl protein levels were determined in two independent *dPANK/fbl*-RNAi strains (1 and 2), in which *dPANK/fbl*-RNAi was ubiquitously expressed. A wild-type strain was used as a control. An anti-dPANK/Fbl antibody was used to detect protein levels with Western blot analysis on whole fly extracts. Tubulin was used as a loading control.

B Survival rate of adult flies was determined for *dPANK/fbl-RNAi* flies, compared to control flies under normal food conditions and after the supplementation of dichloroacetate (DCA) or pantetheine to the food. The assay is explained and visualized in Appendix Fig S3. *dPANK/fbl-RNAi* flies show a decreased survival rate compared to control flies. DCA and pantetheine when added to the food increased viability of *dPANK/fbl-RNAi* flies. Total number of flies used for each group is indicated above each bar. Statistical analyses were carried out using Fisher's exact test.

Source data are available online for this figure.

embryogenesis wing imaginal discs are formed, growing in size during later larval stages and finally forming adult wings during metamorphosis (Appendix Fig S4). Specific parts of the imaginal wing discs develop into specific parts of the adult wing. The larval hinge develops into the adult hinge and the larval wing pouch develops into the adult wing blade forming the majority of the wing structure (Appendix Fig S4; Diaz de la Loza & Thompson, 2017). Disruption of specific genes in parts of the wing during development leads to macroscopically visible wing abnormalities in the adult fly that can be readily scored. A plethora of mutant adult wing phenotypes have been reported, including holes, notches, vein abnormalities, blisters, increased/decreased size, and several others of the wing blade, depending on the specific gene that has been targeted by the RNAi (Bejarano *et al*, 2012).

We performed wing-specific RNAi-mediated knockdown of several components of the CoA-mtACP-PDH pathway during the larval stage. For this, we selected *dPANK/fbl* and *dPPCDC/ppcdc* (the first and third enzyme of the CoA biosynthesis pathway (Bosveld *et al*, 2008) and *mtacp* (*Drosophila mtACP*, Ragone *et al*, 1999). For these genes, *Drosophila* lines harboring RNAi constructs were available, and efficient knockdown could be demonstrated using a wing pouch-specific expression driver (Appendix Fig S4). Downregulation of *dPANK/fbl*, *dPPCDC/ppcdc,* or *mtacp* in the developing larval wing discs leads to a clearly distinguishable wing phenotype in the adult flies, characterized by size decrease and fluid-filled wing blisters varying in size (Fig 5A, C, E and H;

Bejarano *et al*, 2012). The severity of the blister phenotype varied between the knockdowns, most likely because of variations in effectivity of the individual RNAi constructs. This is further substantiated because co-expression of the RNAi-enhancing element *dcr2* with the RNAi construct of each of the three genes resulted in a stronger phenotype (Fig 5B, D, F and I). Combined RNAi-mediated knockdown of both *dPANK/fbl* and *dPPCDC/ppcdc* resulted in an increase of the blistering phenotype (Fig 5G) compared to that arising from downregulation of the single constructs alone (Fig 5C and E). This observation is consistent with the idea of incomplete RNAi-mediated knockdown of either component and suggests an additive phenotype. Together, these experiments validate the genetic knockdown system and implicate a common underlying cellular defect along the CoA-mtACP-PDH pathway leading to a similar phenotype, when the individual steps are compromised.

## Genetic and pharmacologic stimulation of the pyruvate dehydrogenase complex rescues wing phenotypes caused by impaired CoA synthesis

The observation that knockdown of the individual genes *dPANK/fbl*, *dPPCDC/ppcdc,* and *mtacp* cause a phenotype with overt similarities is suggestive for the presence of an underlying common cellular defect. To substantiate this further, we investigated whether boosting PDH activity would rescue the wing-blister phenotype induced

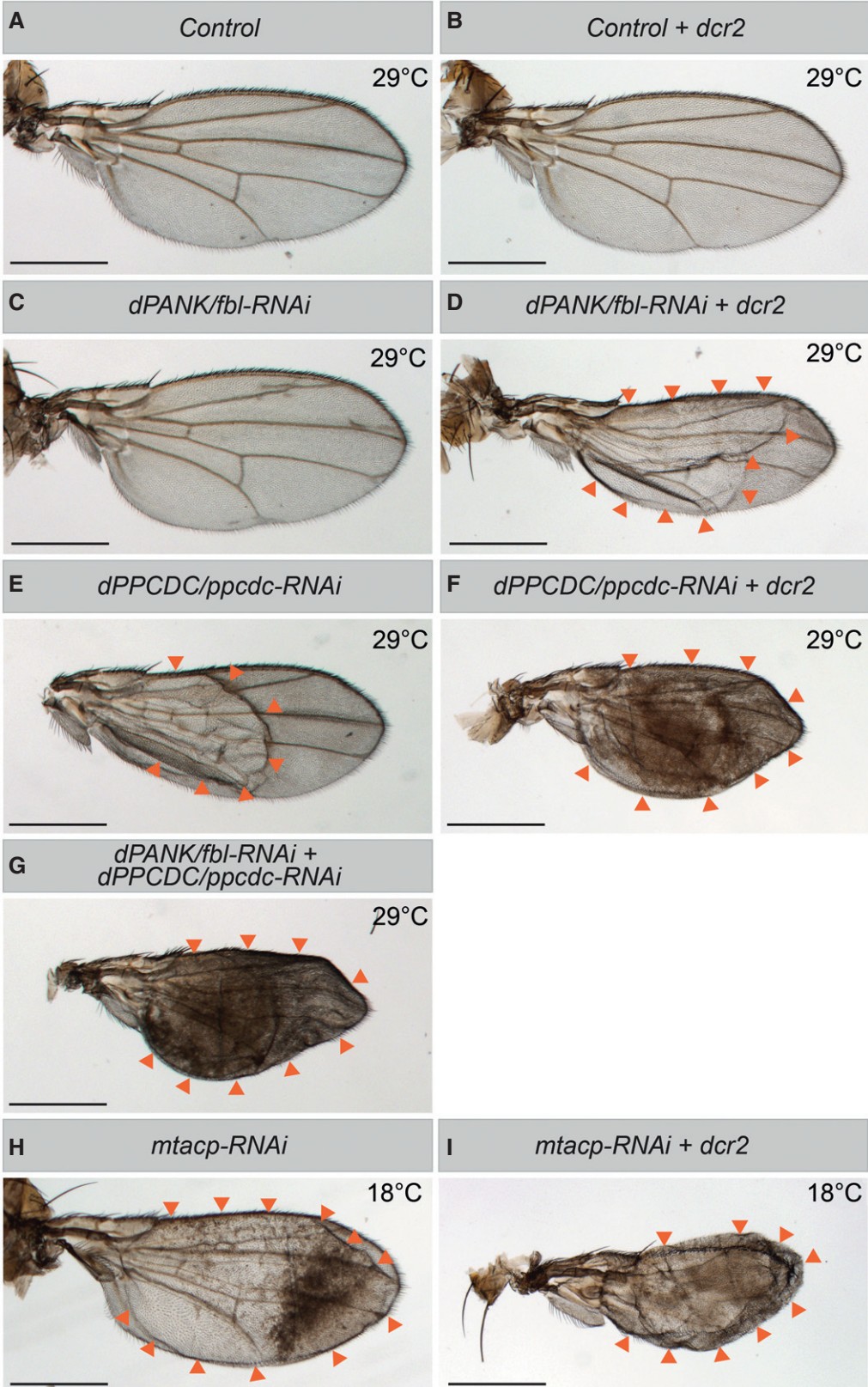

**Figure 5.**

**Figure 5. Downregulation of *dPANK/fbl*, *dPPCDC/ppcdc*, or *mtacp* in the *Drosophila* wing disc leads to blistering in the adult wing.**

*dPANK/fbl*, *dPPCDC/ppcdc*, or *mtacp* was downregulated using gene-specific RNAis during larval development in the wing pouch only (Appendix Fig S4). To enhance the efficiency of the RNAi, experiments were also performed in the presence of the RNAi enhancer *Dicer-2* (*dcr2*). Larvae and the emerging adult flies were raised at 29°C, unless indicated otherwise. The wings of the adult flies were dissected and typical examples of the wings of the adult flies of the indicated genetic backgrounds are provided.

A   Control adult wing.
B   Control adult wing in a *Dicer-2* genetic background.
C   *dPANK/fbl-RNAi* adult wing.
D   *dPANK/fbl-RNAi* adult wing in a *Dicer-2* genetic background.
E   *dPPCDC/ppcdc-RNAi* adult wing
F   *dPPCDC/ppcdc-RNAi* adult wing in a *Dicer-2* genetic background.
G   Combined *dPANK/fbl-RNAi* and *dPPCDC/ppcdc-RNAi* adult wing.
H   *mtacp-RNAi* adult wing. Larvae and flies were raised at 18°C due to decreased viability at 29°C.
I    *mtacp-RNAi* adult wing in a *Dicer-2* genetic background. Larvae and flies were raised at 18°C due to decreased viability at 29°C.

Data information: The wing blisters are visible as brown wrinkled tissue, in contrast to the transparent flat tissue of control wings. Arrow heads were used to mark the perimeter of the blisters, which in some cases does equal the whole wing (F, G, I). Scale bars = 500 μm.

by impaired CoA biosynthesis, comparable to the improved viability we showed in the *dPANK/fbl-RNAi* flies. To investigate this we used the *Drosophila* line with the wing-specific downregulation of *dPPCDC/ppcdc*, in which the blistering phenotype is highly penetrant (in approximately 85% of the adult flies; Fig 6A). We assumed that, in the affected wing, comparable to the HoPAN-treated cells, a fraction of the remaining pool of lipoylated PDH is partly inactivated by the presence of the endogenous PDH-inhibitor pyruvate dehydrogenase kinase (PDK). In addition, PDH can be inactivated by endogenous SIRT4, a lipoamidase that also inhibits PDH (Mathias *et al*, 2014). Therefore, we predicted that residual PDH activity could be increased by interfering chemically or genetically with the action of these inhibitory enzymes (Fig 6B and C). Indeed, we found that feeding the larvae DCA (PDK inhibitor) resulted in a dose-dependent decrease of wing blisters in the adult flies (Fig 6A). In addition, a genetic approach was used (Fig 6C, right part) to decrease the expression of *PDK* and *SIRT4*, by RNAi-mediated knockdown using two different lines per target. The knockdown efficacy of these constructs was verified and they were confirmed to cause no apparent phenotype in control flies (Appendix Fig S5). As predicted, downregulation of these PDH-inhibitors by RNAi in the *dPPCDC/ppcdc*-depleted background partly resolved the wing-blister phenotype (Fig 6D). Non-specific genetic effects were excluded by using an RNAi line, unrelated to this pathway and from the same *Drosophila* RNAi library. This unrelated RNAi line in the *dPPCDC/ppcdc*-depleted background did not affect the blister phenotype (Appendix Fig S6). These results showing that boosting PDH activity can dampen the deleterious effects of impaired CoA biosynthesis and further establish the connection between CoA production and PDH activity.

**Downregulation of *PANK2*, the defective gene in PKAN, leads to a decrease in mtACP in human cells**

Our results in *Drosophila* show that defects in CoA biosynthesis lead to a decrease in *holo*-mtACP levels and decreased lipoylation and activity of PDH, suggesting that a final common pathway through PDH may explain the overlapping phenotypic features of PKAN, CoPAN, MePAN, and PDH-E2 deficiency. Several findings in mammalian systems further support our hypothesis: decreased PDH activity was recently demonstrated in a true PKAN mouse model by Jeong *et al* (2019) and decreased lipoylation of PDH was demonstrated in MePAN patient fibroblasts (Heimer *et al*, 2016). While

these findings support our hypothesis, a key player in this pathway, mtACP has not been investigated in mammalian systems with impaired PANK2 activity. Therefore, we generated HEK293T (human embryonal kidney, Appendix Fig S7) and SH-SY5Y (human neuroblastoma, Appendix Fig S8) cells in which PANK2 protein levels were downregulated. We generated two independent clones per cell line in which, *PANK2*-RNAi expression was inducible by addition of doxycycline and this led to decreased PANK2 protein. Efficiency of PANK2 downregulation was verified using Western blot and qPCR analysis. This demonstrated that PANK2 protein levels were undetectable in both cell lines, whereas PANK1, PANK3, and PANK4 RNA levels were unchanged, establishing the specificity of the PANK2 RNAi constructs (Fig 7A and C, Appendix Figs S9 and S10). Downregulation of the *Drosophila dPANK/fbl* gene is associated with an actual decrease in total CoA levels (Rana *et al*, 2010; Srinivasan *et al*, 2015), explaining the observed decrease in 4′-phosphpantetheinylation of mtACP. Therefore, we sought to determine whether downregulation of *PANK2* in the generated clones led to a measurable decrease in total CoA levels. Total CoA levels were not significantly decreased in any cell lines compared to the controls after 10 days of RNAi treatment despite clear evidence of PANK2 protein loss (Fig 7E and F, Appendix Fig S11). Our results do not exclude the possibility that specific subcellular compartmental CoA levels may be decreased in these cells. Next we determined levels of human mtACP under conditions of downregulation of human *PANK2*. Levels of mtACP were significantly reduced in both PANK2-depleted neuroblastoma cell lines and in one of the two PANK2-depleted HEK293 cell lines compared to controls (Fig 7A–D). Since *apo*-mtACP is unstable and not detectable in other systems, the observed signal for mtACP is most likely the *holo*-(activated form of mtACP; Jackowski & Rock, 1983; Post-Beittenmiller *et al*, 1989). These results show that impaired CoA biosynthesis induced by downregulation of *PANK2* in human cells is associated with a reduction of mtACP which occurs independent of any detectable change in total cellular CoA levels.

## Discussion

Our results demonstrate that impairment of CoA biosynthesis leads to decreased levels of 4′-phosphopantetheinylated *holo*-mtACP, which causes decreased levels of lipoylated, activated PDH.

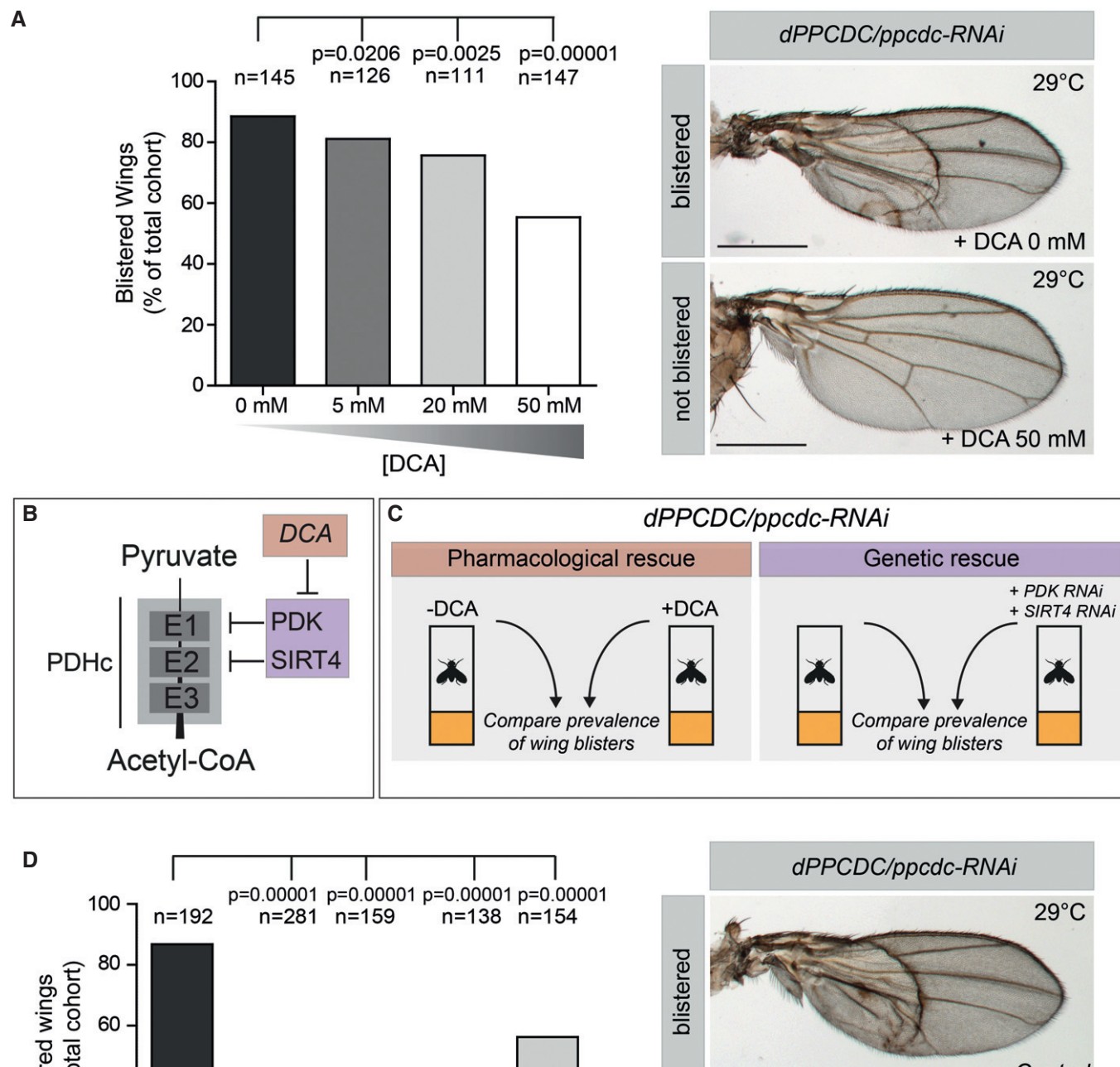

Figure 6.

**Figure 6. Chemical and genetic inhibition of a PDH inactivator rescues the wing phenotypes induced by impaired CoA biosynthesis.**

A Experiments were performed as depicted in (C, left part) and the % of blistered wings was determined in adult *dPPCDC/ppcdc-RNAi* flies raised on control food or on food containing increasing concentrations of DCA. A dose-dependent rescue was observed. Representative images of a blistered and a normal, non-blistered wing are shown. Total number of flies scored for each group over several experiments is indicated above each bar. Statistical analysis was carried out using Fisher's exact test.

B Schematic visualization of the pyruvate dehydrogenase complex (PDHc) and its negative regulators. The PDH complex consists of 3 subunits: PDH-E1, PDH-E2, and PDH-E3. The PDH complex is required for the conversion of pyruvate to an acetyl group, subsequently forming acetyl-CoA. Lipoylation of PDH-E2 is required for normal activation of PDH. Pyruvate dehydrogenase kinase (PDK) phosphorylates PDH-E1 and thereby inactivates PDH. Sirtuin 4 (SIRT4) is a hydrolase abrogating lipoylation of PDH-E2 and thereby also inhibiting PDH activity. Dichloroacetic acid (DCA) inhibits PDK. Inhibiting PDK or SIRT4 expression by RNAi, or inhibiting PDK by DCA is expected to increase activation of PDH.

C Schematic representation of experiments performed to investigate possible pharmacologic and genetic rescue of the adult wing-blister phenotype induced by *dPPCDC/ppcdc-RNAi*. Left part: *dPPCDC/ppcdc-RNAi* flies were raised on control food or on food containing DCA and the number of blistered wings was scored. Right part: Adult wing-blister phenotype of *dPPCDC/ppcdc-RNAi* flies were examined with *dPPCDC/ppcdc-RNAi* flies in which *PDK-RNAi* or *SIRT4-RNAi* were co-expressed, and then the number of blistered wings were scored.

D Experiments were performed as depicted in (C, right part) and the % of adult blistered wings was determined for *dPPCDC/ppcdc-RNAi* flies alone or in a combined background with *PDK-RNAi* or *SIRT4-RNAi*. Two independent RNAi lines were used to downregulate *PDK* and to downregulate *SIRT4*. Co-expression of *RNAi* constructs that targeted PDK or SIRT4 resulted in rescue of the blister phenotype. Total number of flies scored for each group over several experiments is indicated above each bar. Statistical analysis was carried out using Fisher's exact test.

Data information: Scale bars in (A, D) = 500 μm.

Stimulation of PDH activity in the setting of CoA dyshomeostasis rescued phenotypes ranging from viability to organ development, suggesting that impaired PDH function is at least partly responsible for the observed phenotypes. Our results implicate the linear pathway proposed in Fig 2 to explain why impairment at individual steps along this pathway results in overlapping phenotypes in both the *Drosophila* wing model and in the human disorders PKAN, CoPAN, MePAN, and PDH-E2 deficiency.

Here we show a specific and direct effect of impaired CoA biosynthesis on PDH activity. Other enzyme complexes (αKGDH, BCKDH, and GCV) that are regulated by lipoylation (Cronan, 2016) are presumably affected as well, as indicated by decreased levels of pan-lipoylated protein levels (Fig 3D, Appendix Fig S1). The rescue by DCA and SIRT4 downregulation in *Drosophila* under CoA-compromised conditions indicates that at least some of the defects can be attributed to reduced PDH activity because boosting PDH activity causes specific defects to revert to wild-type phenotypes. SIRT4 regulates PDH activity via its lipoamidase activity. This activity of SIRT4 cleaves the lipoyl moiety from the E2 component of PDH and inhibits PDH activity (Mathias *et al*, 2014) explaining the observed rescue. In addition, SIRT4 also interacts with αKGDH-E2 and BCKDH-E2 (Mathias *et al*, 2014). Therefore, it could be possible that SIRT4 can negatively influence lipoylation of these other enzymes as well. Therefore inhibition of SIRT4 would increase lipoylation of αKGDH-E2, and BCKDH-E2 as well as of PDH-E2. Therefore recovery of activity of one or more of these enzymes is likely to explain the potent rescue observed after SIRT4 downregulation.

This rescue is somewhat surprising because boosting these enzyme activities still does not resolve decreased CoA levels or other effects of decreased levels of *holo*-mtACP. mtACP serves numerous functions that are conserved across species. Indeed, NDUFAB1, the human ortholog of mtACP, is a subunit of and required for the assembly of complex I (Vinothkumar *et al*, 2014; Van Vranken *et al*, 2018) and is involved in iron-sulfur biogenesis (Van Vranken *et al*, 2016). It is possible that the majority of the RNAi-induced phenotype in the *Drosophila* wing arises from reduced PDH activity and not from other affected processes downstream from *holo*-mtACP.

Our results demonstrate that, under conditions of decreased levels of total CoA such as in *dPANK/fbl*-downregulated S2 cells or HoPAN-treated samples, levels of *holo*-mtACP are decreased. This result is expected because CoA is the source for the 4′-phosphopantetheinylation and, by reducing its levels, the rate of posttranslational modification that depends on this source would be expected to decrease. A decrease in mtACP levels, most likely reflecting a decrease in active 4′-phosphopantetheinylated *holo*-mtACP, under conditions of normal total CoA levels, as in the *PANK2*-depleted human cells (Fig 7), is more challenging to explain. It is possible that in subcellular compartments like mitochondria, which contain PANK2 and COASY, CoA levels are decreased, inducing a reduction of *holo*-mtACP. It also may be that *PANK2* depletion "only" lowers the rate of CoA *de novo* biosynthesis, which could trigger a compensatory mechanism inhibiting the formation of *holo*-mtACP since this process consumes CoA. The reduction of a consumptive process could explain the decrease in *holo*-mtACP and also could be reconciled with the results showing that total CoA levels remain constant. The prediction would be that under conditions of normal CoA levels but reduced *holo-mtACP,* complex I activity and iron-sulfur cluster formation would be decreased, which is in agreement with the results of Jeong *et al* in their Pank2-depleted mouse model. This further suggests that PKAN, and possibly also CoPAN, is caused by a reduced rate of CoA biosynthesis and not by an actual decrease in CoA levels. This is consistent with the findings that total CoA levels in CoPAN patient fibroblasts are also not reduced (Dusi *et al*, 2014). It is possible then that addition of 4′-phosphopantetheine to Pank2-depleted cells boosts the rate of CoA biosynthesis, thereby explaining its rescuing potential, as reported by Jeong *et al* (2019).

Our hypothesis can explain some of the phenotypic similarities and differences between PKAN, CoPAN, MePAN, and PDH-E2 deficiency. All these disorders selectively damage the basal ganglia, with most of the pathology localizing in globus pallidus. In PKAN and CoPAN, iron accumulates in this structure, but this phenomenon is not observed in the other two disorders (Fig 8). This difference can be explained by postulating that the iron accumulation stems from dysregulation of an intermediate downstream of PANK and COASY but upstream of MECR and PDH-E2. A candidate intermediate is again *holo*-mtACP; it was recently shown in eukaryotic cells that *holo*-ACP is involved in iron-sulfur cluster biogenesis and stability, highlighting a crucial additional role for the 4′-phosphopantetheine–conjugated mtACP (Van Vranken *et al*, 2016;

**A**

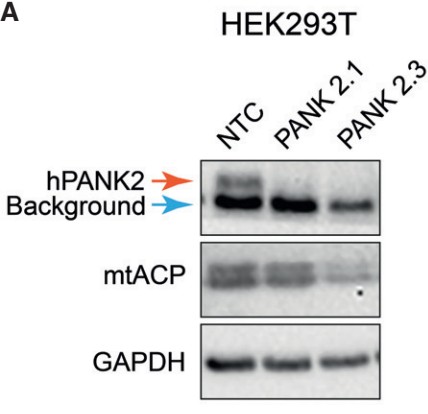

HEK293T

**B**

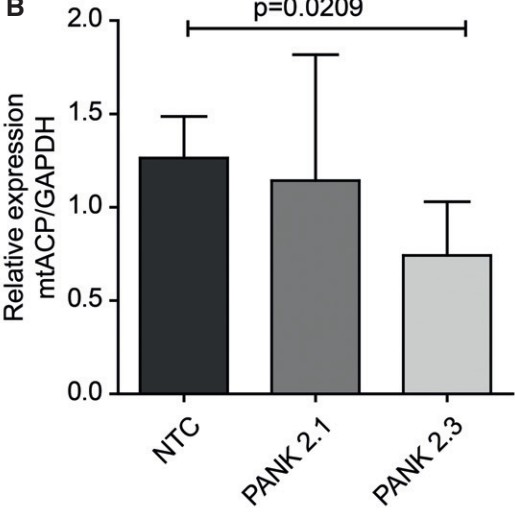

**C**

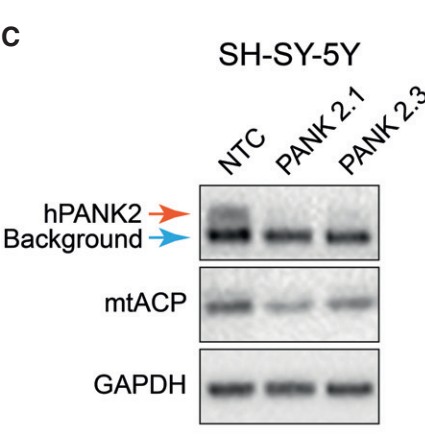

SH-SY-5Y

**D**

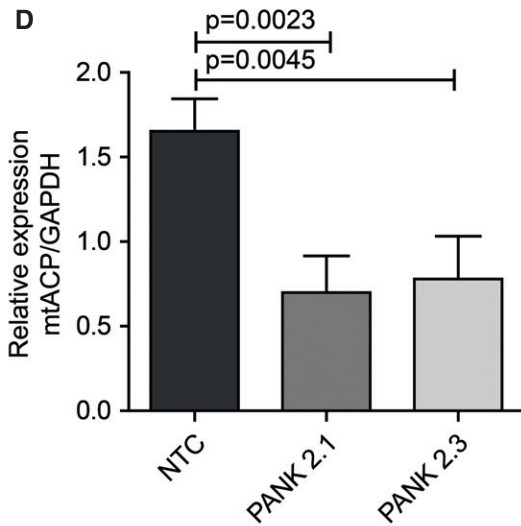

**E**

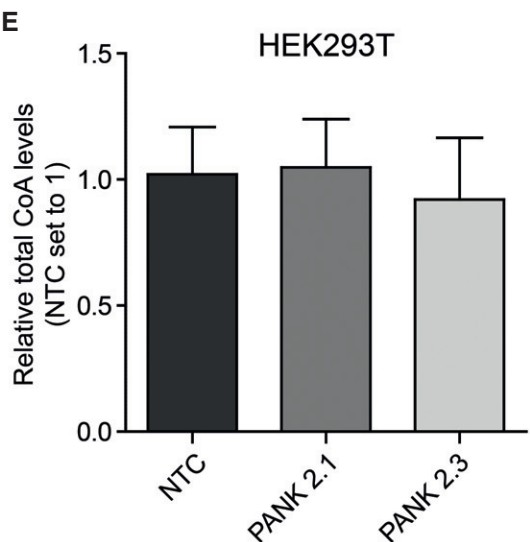

**F**

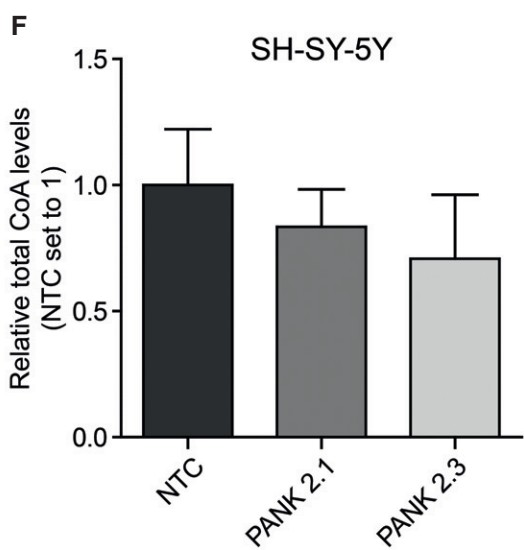

Figure 7.

**Figure 7. Decreased levels of PANK2 are associated with decreased levels of mtACP in mammalian cells.**

A   Western blot analysis of HEK293T cells cultured in vitamin B5-deprived medium. Samples of doxycycline-treated non-targeted controls (NTC) and inducible PANK2 knockout lines, PANK2.1 and PANK2.3, were run and probed for hPANK2 (the hPANK2-specific band is marked by a red arrow, a closely migrating background band is marked by a blue arrow; see also Appendix Fig S10) and for mtACP. GAPDH was used as a control. A significant decrease in mtACP levels was detected in clone 2.3.
B   Quantification of protein band intensities of Westerns shown in (A), performed with Image Studio Lite (LI-COR) and plotted as relative ratio of mtACP to GAPDH. Mean ± SD is given. One-tailed, unpaired Student's $t$-test was performed to compare indicated subsets. $n = 3$ for all samples.
C   As in (A) for SH-SY-5Y cells. A decrease in mtACP levels was detected in both hPANK2 depleted SH-SY-5Y clones.
D   Quantification of protein band intensities of Westerns shown in (C), performed with Image Studio Light (LI-COR) and plotted as relative ratio of mtACP to GAPDH. Mean ± SD is given. One-tailed, unpaired Student's $t$-test was performed to compare indicated subsets. $n = 3$ for all samples.
E, F   Total CoA levels were measured using HPLC as described in Srinivasan et al (2015) in the indicated clones of the HEK293T cells (E) and SH-SY-5Y cells (F) after 10 days of RNAi treatment. HEK293T and SH-SY-5Y cells were cultured in vitamin B5-deprived medium. Mean ± SD is given. $n = 3$ for all samples.

Source data are available online for this figure.

Cory et al, 2017). In *Saccharomyces cerevisiae*, loss of mtACP leads to reduced iron-sulfur cluster formation, inactivation of iron-sulfur cluster-dependent enzymes such as aconitase, and activation of iron-responsive factors Aft1 and Aft2 (Van Vranken et al, 2016). Consistently decreased iron-sulfur cluster levels result in mitochondrial iron overload (Chen et al, 2002). Abnormal iron homeostasis and reduced aconitase activity are characteristics of PKAN patient fibroblasts, as well as IPSC-derived neurons (Santambrogio et al, 2015; Orellana et al, 2016). Iron dyshomeostasis and reduced aconitase activity are also reported by Jeong et al in their PKAN mouse model (Jeong et al, 2019). This explains why mitochondrial iron dyshomeostasis and accumulation resulting from mtACP dysfunction is observed in PKAN and CoPAN, diseases associated with steps upstream of *holo*-mtACP, but not in MePAN and PDH-E2 deficiencies, diseases associated with steps downstream of *holo*-mtACP. Consistent with this, we predict that MECR deficiency compromises lipoic acid production without affecting mtACP levels or activity (Fig 8). Recently, Klopstock et al (2019) reported that iron chelation is effective in reducing the brain iron content in PKAN patients; however, the pathway proposed here would predict that iron chelation therapy alone would be insufficient to fully counteract neurodegeneration in patients with PKAN or CoPAN.

Our results and models complement and agree with the work of Jeong et al and underscore the clinical relevance of our findings. Jeong et al demonstrate, in a mouse model of PKAN, the presence of a specific set of perturbations in the globus pallidus. These alterations include: impaired complex I function with decreased oxidative phosphorylation, impaired lipoic acid production with loss of activity of a lipoylated enzyme (PDH), and iron dyshomeostasis with loss of activity of dependent enzymes and processes

presumably from impaired iron-sulfur cluster biogenesis. All their reported findings are consistent with a primary defect in 4′-phospho-pantetheinylated mtACP, including their observation of rescue of all molecular changes in mouse globus pallidus by administration of 4′-phosphopantetheine.

Despite providing a compelling hypothesis and strong results to support it, we recognize that the *Drosophila* model does not reflect the full complexity of the mammalian system, especially regarding the metabolic step compromised in PKAN. Downregulation of the single *dPANK/fbl* gene in fruit flies evokes a more severe phenotype compared to downregulation of only PANK2 in mammalian systems. Alternative hypotheses to explain decreased activity of PDH from a CoA metabolic defect should also be considered. One attractive hypothesis is based on the process of CoAlation, a recently identified posttranslational modification. A range of proteins was identified, including PDK, that can be reversibly modified by covalent attachment of CoA, which influences their activity. PDK activity is inhibited by CoAlation (Tsuchiya et al, 2017). Therefore, an alternative explanation for our results may be that under conditions of impaired CoA biosynthesis, levels of PDK-CoAlation are reduced, leading to increased activity of PDK and therefore decreased activity of PDH.

Our hypothesis does not explain why the globus pallidus is selectively damaged in the four diseases. For PKAN, it may be that the other PANKs compensate for the loss of PANK2 in areas outside the globus pallidus; however, this redundancy does not exist for CoPAN, MePAN, and PDH-E2 deficiency. Our hypothesis suggests that the globus pallidus is vulnerable to decreased activity of PDH; however, an explanation for this sensitivity is still lacking. The PKAN mouse model reported by Jeong et al (2019) is a powerful

**Figure 8. Phenotypic features of PKAN, CoPAN, MePAN, and PDH-E2 deficiency and their hierarchy on a pathophysiological axis.**

Left part: The names of the diseases PKAN, CoPAN, MePAN, and PDH-E2 deficiency are given in full with their corresponding OMIM numbers, clinical features, and schematics of representative MR images (T2-weighted). Schematic drawing is provided to identify the relevant basal ganglia structures in healthy brain, including putamen, caudate nucleus, and globus pallidi. In healthy brain, these structures appear isointense on T2-weighted imaging to surrounding gray matter until early adulthood; in all four diseases, T2-hyperintense signal is seen in the globus pallidus bilaterally (McWilliam et al, 2010; Kruer et al, 2012; Dusi et al, 2014; Heimer et al, 2016), and in PKAN and CoPAN, pallidal signal hypointensity is also seen, which is indicative of high iron levels (indicated with an arrowhead). [Note that T2-hyperintense putamina were also been observed in some MePAN patients (Heimer et al, 2016)].The pathway on the right of the figure consolidates the proposed biochemical linkages, explaining how mutations in four different genes all result in PDH-E2 deficiency and thus lead to shared phenotypic features. From top to bottom: pantothenate kinase (PANK), phosphopantothenoylcysteine synthetase (PPCS), phosphopantothenoylcysteine decarboxylase (PPCDC), and coenzyme A synthase (COASY) are enzymes required for the *de novo* biosynthesis of CoA. Mitochondrial acyl carrier protein (mtACP) undergoes posttranslational modification, gaining a 4′-phosphopantetheine moiety that is derived from CoA in order to form the active *holo*-mtACP. *Holo*-mtACP is required for lipoylation of PDH-E2, a requirement for activation of the PDH complex. We propose that a decrease in CoA biosynthesis leads to decreased amounts of *holo*-mtACP, decreased lipoylation of PDH-E2, and decreased activity of PDH. *holo*-mtACP is also required for the biogenesis of iron-sulfur clusters. Impaired iron-sulfur cluster formation leads to iron dyshomeostasis. Therefore, this model also explains iron accumulation in diseases associated with defects upstream of *holo*-mtACP but not in those downstream of *holo*-mtACP. Arrowhead indicates iron accumulation in schematic drawing of PKAN and CoPAN.

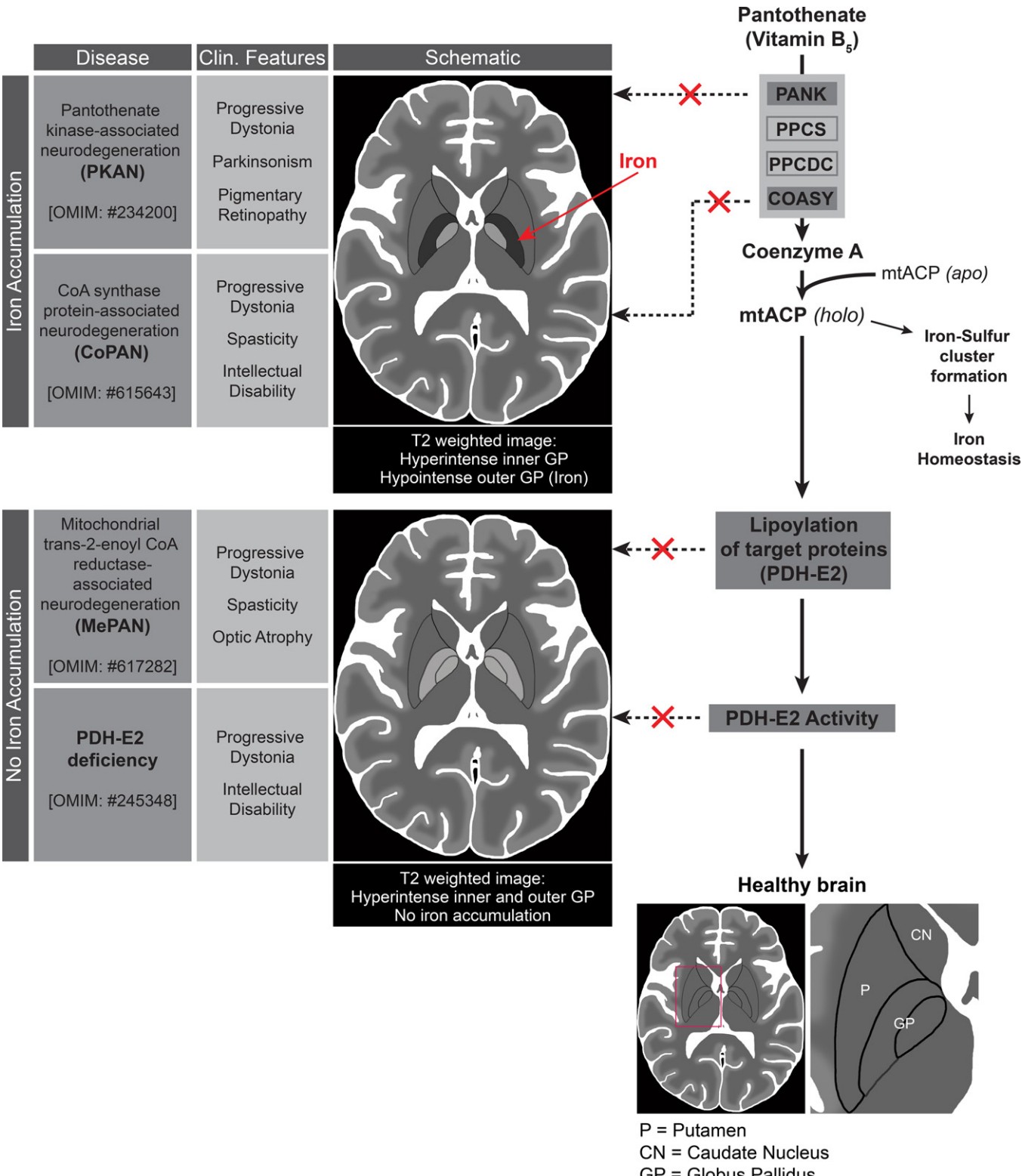

**Figure 8.**

tool that may enable discovery of an answer to this important question.

Our model presented in Fig 8, in combination with the results provided by Jeong *et al* (2019) and Klopstock *et al* (2019), provides suggestions for possible therapies for the four neurodegenerative diseases. DCA is a blood–brain barrierpermeable drug (Abemayor *et al*, 1984; Kuroda *et al*, 1984) that has been investigated as a candidate therapeutic for metabolic diseases as well as cancer (Stacpoole *et al*, 1992; Stacpoole *et al*, 2006; Michelakis *et al*, 2010; Abdelmalak *et al*, 2013). Despite achieving biochemical remission of lactic acidosis in a genetically heterogeneous cohort of children with congenital lactic acidosis, DCA failed to slow neurological decline (Stacpoole *et al*, 2006). Moreover, while DCA rescues the *Drosophila* wing model, it would most likely not rescue iron dyshomeostasis or other dysfunction arising downstream of *holo*-mtACP such as of aconitase and complex I activities in PKAN and CoPAN patients. Based on the results by Jeong *et al*, 4′-phosphopantetheine is the most promising therapeutic strategy for PKAN because it restores an early step in the affected pathway and therefore would be predicted to prevent all downstream impaired processes. In CoPAN patients, a CoA biosynthesis step is affected downstream of 4′-phosphopantetheine; therefore, we would not expect 4′-phosphopantetheine to prevent disease changes in CoPAN. However, based on our proposed pathway, an iron chelator in combination with DCA or a SIRT4 inhibitor may be beneficial in CoPAN. Finally, DCA or a SIRT4 inhibitor may also be beneficial in MePAN and to some extend in PDH-E2 deficiency, because DCA addition will not rescue severe loss of function mutations. Additional investigations in mammalian models are required to substantiate these ideas.

# Materials and Methods

### S2 cell culture, transfection, HOPAN, and CoA treatment

*Drosophila* Schneider's S2 cells were maintained at standard conditions as described previously (Srinivasan *et al*, 2015). Here, cells in their exponential phase of growth were transfected (Effectene, Qiagen) with the mtACP WT or mutant constructs listed below and grown for 2 days.

HoPan and CoA or DCA treatment were done on S2 cells in their exponential phase and treated with 0.5 mM HoPan in the presence or absence of 100 μM CoA or 10 mM DCA. Cells were treated for 2, 4, or 7 days, and untreated S2 cells were used as control.

### Cloning of *mtACP* (mutant) constructs

In order to overexpress wild-type or mutant *mtACP*, constructs were created in the following manner. An *mtACP* cDNA clone for isoform B was obtained from the *Drosophila* Genomics Resource Center (AT22870; FBcl0025645) and multiplied by PCR using primers flanked by EcoRI and XhoI restriction sites:

The pAc5.1 vector (Invitrogen) was digested using EcoRI and XhoI and ligated with the *mtacp* PCR product. Competent cells were transfected with the ligated construct and the purified construct was sequenced to ensure its fidelity. Constructs overexpressing mutant *mtacp* were subsequently created by site-directed mutagenesis of

this construct using mutagenesis primers (Q5 Site-directed Mutagenesis Kit, New England Biolabs). The fidelity of the resulting constructs was verified by sequencing. Primers sequences are presented in Appendix Table S1.

### Lentiviral transductions

Inducible lentiviral shRNA vectors targeting a non-targeted control (NTC) or hPANK2 were obtained from Dharmacon (for sequences, see supplementary data) and lentivirus was produced as previously described (Schepers *et al*, 2007). 293T cells (ATCC CRL-3216) and SH-SY5Y cells (ATCC CRL-2266) were transduced in 2 consecutive rounds of 8–12 h with lentiviral supernatant supplemented with 10% FCS and polybrene (4 μg/ml; Sigma). Transduced 293T and SH-SY5Y cells were selected in a medium containing 0.6 μg/ml or 0.2 μg/ml puromycin (Thermo Fischer Scientific), respectively, for 1 week.

### FACS analysis

After 1 week of puromycin selection, expression of the microRNA-TurboGFP cassette was induced with doxycycline (Sigma, concentrations ranging from 0 to 1.5 μg/ml) for 3 days and analyzed for TurboGFP expression on an LSRII (Becton Dickinson) flow cytometer, and data were analyzed using FlowJo software (FlowJo V10).

### Inducible hPANK2 knockdown cells

Cells were produced as described above. HEK293T and SH-SY5Y inducible hPank2 KD cells were maintained in Dulbecco's Modified Eagle's Medium (DMEM, Sigma) supplemented with 10% fetal bovine serum (FBS, Greiner Bio-one) and antibiotics (penicillin/streptomycin, Invitrogen) in 5% $CO_2$ at 37°C. Induction of the microRNA-TurboGFP in SH-SY5Y was done with 0.5 μg/ml doxycycline for 7 days. Induction of the microRNA-TurboGFP in HEK293T was done with 1 μg/ml doxycycline for 14 days, in custom made DMEM without vitamin B5 (Thermo Scientific) supplemented with dialyzed FBS (Thermo Scientific) and antibiotics. HoPan treatment (0.25 mM) was done from day 7 till day 14.

A complete list of vectors used, commercially available cell lines, and lines created in this publication can be found in Appendix Tables S1 and S2. The cell lines were regularly visually inspected and tested negative for mycoplasma contamination.

### Western blot analysis and antibodies

For Western blot analysis, cells were dissolved in 2× Laemli buffer, sonicated, and boiled for 5 min with 5% β-mercaptoethanol (Sigma). Protein concentration was determined using DC protein assay (Bio-Rad). Equal amounts of protein (10–30 μg) were loaded on a 10, 12, or 4–20% gradient gel (Bio-Rad), and transferred onto PVDF membranes using the Trans Blot Turbo System (Bio-Rad). Membranes were blocked in 5% fat-free milk for 1 h at room temperature, and then rinsed in PBS-Tween 20. Incubations with primary antibodies were done overnight at 4°C followed by incubations with HRP-conjugated secondary antibodies (Amersham 1:5,000) for 1.5 h at room temperature. Detection was performed using ECL reagent (Thermoscientific) and visualized using the

ChemiDoc imager (Bio-Rad). The following primary antibodies were used: anti-mtACP antibody (Abcam, 1:1,000), anti-lipoic acid (Merck, 1:1,000), anti-PDH-E2 (Abcam, 1:1,000), anti-α-Tubulin (Sigma, 1:5,000), anti-GAPDH (Fitzgerald, 1:10,000), anti-dPank/Fbl (Bosveld *et al*, 2008; 1:1,000), and anti-hPANK2 (Origene 1:500). A complete list of antibodies used can be found in Appendix Table S3.

### PDH activity measurements

S2 cells were cultured as described above for 4 days in control medium or medium containing 500 μM HoPan. Cells were pelleted at approximately $10^6$ cells/pellet and washed once with PBS, and later pyruvate dehydrogenase complex activity was measured using the Pyruvate Dehydrogenase Activity Colorimetric Assay Kit (Cat#K679-100, BioVision) according to manufacturer's instructions. Three biological replicates were used per measurement, with each biological replicate measured in (technical) triplicate. Protein concentration was determined using BCA Protein Assay Kit (ThermoScientific) according to manufacturer's instructions. All measurements were recorded using a VarioSkan Lux plate reader; analysis was performed with GraphPad (see section "Statistical analysis").

### *Drosophila* maintenance and genetics

*Drosophila melanogaster* stocks were maintained on standard cornmeal agar fly food (containing water, agar 17 g/l, sugar 54 g/l, yeast extract 26 g/l, and nipagin 1.3 g/l) at 22°C. Crosses were raised at various temperatures as indicated in the text/legends. The stocks were either obtained from the Bloomington Stock Centre (Indiana University, USA) or the VDRC (Vienna *Drosophila* RNAi Collection, Vienna, Austria).

### Stocks used for this project

A complete list of commercially available fly lines, lines created in the context of this publication, and detailed genotypes can be found in Appendix.

Crosses were raised at various temperatures as indicated in the text/legends.

### Validation of *UAS-dPANK/fbl-RNAi* and *UAS-dPPCDC/ppcdc-RNAi* lines by meiotic recombination

Based on previous results (expression with various drivers, data not shown), we predicted that our *UAS-dPANK/fbl-RNAi* (#101437) and *UAS-dPPCDC/ppcdc-RNAi* (#104495) KK lines, obtained from the VDRC, might contain not only the regular RNAi transgene at location 30B, but an additional one at 40D, which has been shown to act as a phenotypical enhancer and able to cause false-positive results, especially when combined with ubiquitous or wing drivers (Green *et al*, 2014; Vissers *et al*, 2016).

To test those lines for RNAi transgene integration site occupancy, we used genomic DNA preparation, followed by PCR analysis, as described previously (Vissers *et al*, 2016). When both lines were confirmed to carry transgenes at integration sites 30B and 40D, we employed meiotic recombination to remove the one at 40D. Lines with the remaining insertion at 30B were confirmed by PCR (Appendix Fig S1A) and tested for knockdown by

immunohistochemistry (anti-Fbl, Fig 4A) and qPCR (*fbl* and *PPCDC*, Appendix Fig S1B and C). The "30B-only" cleaned-up lines were used for all experiments presented in this publication unless stated specifically otherwise (also see stock list in the Appendix for details).

### Immunofluorescence analysis of *Drosophila* wing imaginal discs

For immunofluorescence of *Drosophila* wing discs, the crosses were raised at 29°C (*MS1096-GAL4; UAS-GFP × UAS-GFP* as control or x *UAS-dPANK/fbl-RNAi*) or 22°C (*MS1096-GAL4; UAS-GFP × UAS-GFP* as control or x *UAS-mtacp-RNAi #29528*). Wandering L3 larvae (day 5) were collected and their wing discs dissected in ice-cold phosphate-buffered saline (PBS). The discs were fixed with 4% formaldehyde (Thermo Scientific Pierce) for 30 min, washed for 3 × 20 min with phosphate-buffered saline (PBS) + 0.1% Triton X-100 (Sigma Aldrich), and afterward incubated in primary antibody rabbit anti-Fbl (Bosveld *et al*, 2008; 1:500) or rabbit anti-mtACP (ThermoFisher PA5-22191, 1:500) in PBS + 0.1% Triton X-100 overnight to visualize the presence/absence/localization of Fbl or mtACP. After an additional washing step of 3 × 20 min in PBS + 0.1% Triton X-100, the discs were incubated in secondary goat anti-rabbit-Alexa488 antibody (Molecular Probes) for 2 h at room temperature. DAPI (0.2 μg/ml; Thermo Scientific) was used to visualize DNA. Finally, the samples were mounted in 80% glycerol and analyzed using a Zeiss-LSM780 NLO confocal microscope with Zeiss software. Adobe Photoshop and Illustrator (Adobe Systems Incorporated, San Jose, California, USA) were used for image assembly. A complete list of antibodies used can be found in Appendix Table S3.

### *Drosophila* survival rate experiments

Virgin female *Actin-GAL4/CyO* flies were crossed with males of either *UAS-GFP* or *UAS-dPANK/fbl-RNAi* at 29°C on standard fly food, or food supplemented with either sodium dichloroacetate (DCA, Sigma) or pantethine (Pan, Sigma) at indicated concentrations. The flies were allowed to lay eggs for 5 days, after which the adults were discarded. The emerging adult (eclosing) male flies were evaluated for the presence or absence of the *CyO* dominant marker. Non-*CyO* flies were selected because only in these flies the *dPANK/fbl-RNAi* (*Actin-GAL4/UAS-dPANK/fbl-RNAi*) or the GFP (*Actin-GAL4/UAS-GFP,* as control) is expressed. Progeny were counted daily over a period of 5 days to evaluate the total survival rate of the adult flies. To determine the survival rate of *dPANK/fbl-RNAi* flies, which is an indicator for the viability, the number of male *Actin-GAL4/UAS-dPANK/fbl-RNAi* flies was divided by the total number of male flies (CyO and non-CyO). At least 6 separate vials containing offspring were used per condition.

### Mounting and imaging of adult fly wings

To image wings of adult flies from various crosses, F1 males or females of the indicated genotypes were collected for a period of 3 days and kept for an additional 2–3 days after eclosion to allow for optimal unfolding and clearance of the wings. Afterward, they were transferred into 70% ethanol and stored for at least 2–3 days. The wings were removed with tweezers, mounted on slides in 80% glycerol, and imaged with a light microscope (Olympus BX-50) at 2× magnification. Adobe Photoshop and Illustrator (Adobe Systems

Incorporated, San Jose, California, USA) were used for visualization.

### Pharmacological and genetic rescue of d*PPCDC/ppcdc-RNAi*-induced wing blisters

For the pharmacological rescue, *MS1096-GAL4, UAS-GFP or MS1096-GAL4, and UAS-dPPCDC/ppcdc-RNAi* females were crossed with *UAS-GFP* males at 29°C on standard fly food, or food supplemented with sodium dichloroacetate (DCA, Sigma) at indicated concentrations.

For the genetic rescue, *MS1096-GAL4, UAS-GFP or MS1096-GAL4, and UAS-dPPCDC/ppcdc-RNAi* females were crossed with *UAS-GFP, UAS-PDK-RNAi (#28635 and #35142),* or *UAS-SIRT4-RNAi (#33984 and #36588)* males at 29°C on standard fly food.

The females were allowed to lay eggs for 5 days, after which the parents were discarded. F1 males were collected and imaged as described above and analyzed for the presence or absence of "blisters".

The adult *Drosophila* wing consists of two layers, a dorsal (upper) and a ventral (lower) one, which are tightly connected to each other (by adhesion molecules). "Blistering" of the wing has been described in general as a condition in which dorsal and ventral wing surfaces separate to form a hemolymph-filled blister (Martin *et al*, 1999). After mounting and imaging, they appear transparent and the remains of the blister are only detectable by their crinkled surface (Fig 5D and E). In blisters that encompass the whole wing, hemolymph (and associated cell debris) is not cleared sufficiently and after mounting and imaging these wings appear inflated with brownish discolorations (Fig 5F–I).

Arrow heads were used to mark the perimeter of the blisters, which in some cases does equal the whole wing. In comparison, wings were scored as "not blistered", if they appeared flat, structurally normal, and/or no discoloration was observed (Fig 5A and B). For visualization and quantification, images were recorded, using Adobe Photoshop and Illustrator (Adobe Systems Incorporated, San Jose, California, USA) and GraphPad Prism software (GraphPad Software, San Diego, CA, USA). For randomly chosen experiments, the amount of blisters was counted by independent researchers. Blinding was not performed.

### qPCR

For quantitative real-time PCR (qPCR), we collected 8 specimens from fly samples, in most cases adult flies, from crosses of *Actin-GAL4* with *UAS-dPANK/fbl-RNAi F10 and F20, UAS-dPPCDC/ppcdc-RNAi P7 and P17, UAS-PDK-RNAi #28635 and #35142,* and *UAS-SIRT4-RNAi #333984 and #36588*. The cross of *Actin-GAL4* with both *UAS-dPPCDC/ppcdc-RNAis* results in mid- to late-pupal lethality, so we collected those samples during pupal development. The samples were crushed with a motor pestle, lysed in Trizol (Life Technologies), and RNA isolation was performed according to standard protocol. The isolated RNA was treated with Turbo DNAse (Ambion) followed by addition of random primers and reverse transcription using M-MLV (Invitrogen). Quantitative PCR was performed using Sybr Green (Bio-Rad) on a CFX Connect Real Time System (Bio-Rad). Quantitative PCR was performed using primer sets to detect *dPANK/fbl*, *dPPCDC/ppcdc*, *PDK,* and *SIRT4* mRNA

expression, and normalized against *Rp49* mRNA-loading controls. qPCR was performed in the human cells to determine PANK1-4 levels. ells were collected and lysed in RLT and RNA isolation was performed using the RNeasy kit from Qiagen according to the manufacturer's instructions. Fly samples were normalized for rp49, and human samples were normalized for B2M and UBC.

A complete list of qPCR primers can be found in Appendix Table S2.

### Statistical analysis

#### Densitometry and statistical analysis
Western blot analyses were measured using Image Studio Lite (Li-Cor, www.licor.com/bio/image-studio-lite). Statistical analyses were performed using Excel 2016 or GraphPad PRISM 5 (San Diego, CA, USA), with either unpaired Student's *t*-test or Fisher's exact test. Statistically significant results are depicted within each figure. Unless stated specifically in the figure legend, all the data were presented as mean ± SD.

For proportional data, power calculations for comparing proportions were used. *F*-tests were used to test if variances of the tested populations were equal. The number of flies assured a statistical power of 0.8 with a confidence level of 0.95 for Fisher's exact test and Student's *t*-tests (proportional power calculations; Wang & Chow, 2007), taking into account that the expected proportion of responders is around 25–80%.

**Expanded View** for this article is available online.

### Acknowledgements
We thank the TRiP at Harvard Medical School (NIH/NIGMS R01-GM084947), the Bloomington Stock Center, and the VDRC for providing fly stocks and transgenic RNAi lines used in this study. We thank Klary Niezen for technical assistance and J. Kalervo Hiltunen and Alexander J. Kastaniotis for critical reading of the manuscript. This work was supported by a VICI grant to O.S. (NWO-grant 865.10.012). Part of the work has been performed at the UMCG Microscopy and Imaging Center (UMIC), which is sponsored by NWO-grant 175-010-2009-023. MAVL received a PhD stipendium from UMCG. The authors declare that they have no conflict of interest other than stated below. KJA is supported by grants from the Academy of Finland (267388) and the Sigrid Juselius Foundation.

### Author contributions
RAL, YY, HS, BMB, MAV-L, MAT, SJH, NAG, and OCMS designed the research studies, analyzed data, and wrote the manuscript. RAL, YY, HS, NAG, MAV-L, KJA, MAT, and MvdZ conducted experiments, and acquired and analyzed data. KJA acquired samples. MvdZ, RAL, and HS generated essential reagents.

### Conflict of interest
Dr. Hayflick and Dr. Sibon are co-inventors on a patent application for 4'-phosphopantetheine for use in disorders exclusive of PKAN. Dr. Sibon is a co-inventor on a patent application for acetyl-4'-phosphopantetheine for use in PKAN and in related disorders. Dr. Hayflick is a non-compensated member of the Scientific Advisory Board of BioPontis Alliance, a non-profit organization. Dr. Hayflick is a non-compensated member of the Scientific and Medical Advisory Board of the NBIA Disorders Association, a non-profit lay advocacy organization. Dr. Hayflick is a non-compensated member of the Scientific and Medical Advisory Board of the NBIA Alliance, a non-profit lay advocacy

## The paper explained

### Problem

PKAN, CoPAN, MePAN, and PDH-E2 deficiency are neurodegenerative diseases caused by mutations in four different genes for enzymes important in distinct metabolic pathways. The four diseases selectively damage the basal ganglia, which suggests that they share a common metabolic pathogenesis. Identifying this shared metabolic defect is essential to understanding their pathophysiology and designing effective therapeutics for all the four diseases.

### Results

Fruitfly and mammalian cell systems were used to model the diseases and reveal a shared metabolic pathway. We present evidence that links the pathways of coenzyme A biosynthesis to the activity of pyruvate dehydrogenase (PDH) via mitochondrial acyl carrier protein. Chemical and genetic strategies were employed to boost PDH activity, which is a final common defect in the four disorders, leading to correction of defects observed in our disease models.

### Impact

Our results provide an explanation for the overlap in clinical phenotypes of PKAN, CoPAN, MePAN, and PDH-E2 deficiency and suggest treatment strategies for all four diseases.

organization. Dr. Hayflick serves as non-compensated executive for the Spoonbill Foundation, a not-for-profit organization that may benefit from the results of this research and technology. Dr. Sibon serves as non-compensated executive for the Stichting Lepelaar, a not-for-profit organization that may benefit from the results of this research and technology. This potential conflict of interest has been reviewed and managed by OHSU and the UMCG.

## For more information

(i)   http://flybase.org/
(ii)  https://www.omim.org/entry/234200
(iii) https://www.omim.org/entry/615643
(iv)  https://www.omim.org/entry/617282
(v)   https://www.omim.org/entry/245348

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
