## [Review Process File · EMBO Molecular Medicine]

CoA-dependent activation of mitochondrial acyl carrier protein links four neurodegenerative diseases

Roald A. Lambrechts, Hein Schepers, Yi Yu, Marianne van der Zwaag, Kaija J. Autio, Marcel A. Vieira-Lara, Barbara M. Bakker, Marina A. Tijssen, Susan J. Hayflick, Nicola A. Grzeschik, Ody C.M. Sibon

Review timeline:

Submission date:	5 March 2019
Editorial Decision:	10 April 2019
Revision received:	22 August 2019
Editorial Decision:	1 October 2019
Revision received:	11 October 2019
Accepted:	15 October 2019

Editor: Céline Carret

Transaction Report:

1st Editorial Decision

10 April 2019

Thank you for the submission of your manuscript to EMBO Molecular Medicine. We have now heard back from the two referees whom we asked to evaluate your manuscript.

You will see from the comments pasted below that both referees enjoyed the article and have similar and overlapping concerns. The fly model remains a fly model in that it doesn't cover the mammalian complexity and by not investigating in other directions than the PDH enzyme, you are restricting the conclusiveness of the data and somehow weakening your study as compared to the accompanying article (see below). Further, the study needs to be considerably improved in terms of clarity, providing details and explanations and a better discussion along with high resolution WB.

Following our cross-commenting exercise, ref 2 says: "explaining (if they can) why PDH deserves to be the singular focus of their studies, and potential future treatment are the key points to bring this manuscript closer to being acceptable for publication." This referee also notes that "attaching [their paper] to the Hayflick manuscript may actually reduce its impact, as it would highlight the limitations of the fly model (compared to a mammalian one) and also highlight the complexity of the defect's effects, specifically that it involves much more than just PDH."

Therefore, I would like to invite you to revise your article and would encourage you to add more experimental evidence to strengthen the findings as much as you can. I believe that both articles strengthen one another, so would advise against dissociation. But therefore, further data are needed to bring your article to that level.

Please note that EMBO Molecular Medicine strongly supports a single round of revision and that, as acceptance or rejection of the manuscript will depend on another round of review, your responses should be as complete as possible.

Please also contact us as soon as possible if similar work is published elsewhere. If other work is published, we may not be able to extend the revision period beyond three months.

I look forward to receiving your revised manuscript.

***** Reviewer's comments *****

Referee #1 (Comments on Novelty/Model System for Author):

The fly model system is well established, but the authors did not explain their interpretations well or describe the overall biological pathway clearly.

Referee #1 (Remarks for Author):

This paper was submitted at the same time as a paper by Jeong et al. It is doubtful that this paper could stand alone at the moment, because it is unclearly written, and much of the terminology is directed to a fly audience, and the biochemical pathways are poorly described, and they completely ignored the potential role of 3 other enzyme systems in which lipoylation is required. It seems unlikely that their efforts to modify PDH could compensate, when alpha keto dehydrogenase, the glycine cleavage system, and the branched chain amino acid pathways are presumably also compromised.

There are many areas that need improvement:

1. the abstract is confusing and potentially misleading. What does the sentence "impaired CoA homeostasis leads to decreased 4 phosphopant... really mean? Their point in most experiments is that 4 phosphopant.. is required for CoA synthesis. the wording is circular. Also, by activation the "most downstream" is an unsupported assertion. What about the TCA cycle component and the other two active enzyme systems that require lipoylation? Aren't they more downstream? were they affected?
2. They never explained the difference between de novo FA synthesis in mitochondria vs. FA consumption. A diagram and explanation with the full names of the enzymes and relevant intermediates would help here.
- 3 They should have included a reference and explanation for the role of LIAS and included a strong reference, such as one by Squire Booker.
4. In figure 2, they did not explain the relevance and meaning of COO enclosed in a red circle. In general, the figure legends needed a stronger narrative and better presentation.
5. In a full section of the paper, they seem to think that PDH is the only enzyme that should be discussed and might contribute to the phenotype. They need to discuss this with more intellectual rigor.
6. In their fly kos, they need to convert to language that molecular medicine readers can easily follow. Instead of writing out UAS GFP etc. they need to point out which lines have a knockout of which gene in language accessible to most biologists. The full explanation can be included in the M&M,
7. The notion that hypoxia would lead to the pathophysiology is not correctly represented. It would be that oxygen availability is normal, but respiration is impaired. The reviewer did not have time to review that paper, but confusing hypoxia with impaired respiration is a common mistake, and it needs to be clarified here.
8. Should specify that activation of iron responsive factors was in yeast model system and name them.
9. Does the final sentence contradict the conclusions of the companion paper. No disease modifying treatments currently available seems inappropriate since the other paper described a strong candidate.
10. Better description of figure 4 needed- what does the box demonstrate?
11. Figure 6, the western of hPANK2 is not of acceptable quality. There is a huge apparently non-specific band. It is possible to do better mammalian westerns for PANK2

12. The assertion that mtACP is done in the two PANK2 kd lines is not strong. Better data and more experimental points are needed here.

13. They should have explained the potential significance of hyperintensity in their discussion. Overall, this paper is difficult to read, contains some weak data, and somehow ignores everything that would be affected other than PDH, which is not intellectually sufficiently rigorous for the anticipated reading audience. Some sentences were so confusing that this reviewer gave up. A strong rewrite would make a big difference.

Referee #2 (Comments on Novelty/Model System for Author):

The model is perfectly adequate for testing the questions that the researchers set to answer. However, it is limited in explaining how the results would translate to a mammalian system, as the relevant genetic defect is in a gene that encodes an enzyme (pantothenate kinase/PanK) that occurs as multiple isoforms in humans, of which the relevant PanK localizes to mitochondria. In contrast, *Drosophila* only has a single PanK. This makes for a simpler system that gives rise to the pronounced phenotypes that made the study possible. However, the lack of complexity also makes it impossible to predict how the results would translate to the mammalian system where subcellular differentiation is at play. See review for further comments in this regard.

Referee #2 (Remarks for Author):

In the manuscript by Lambrechts et al. the authors demonstrate a link between levels of the essential cofactor of central energy metabolism, coenzyme A (CoA), and the active (holo) form of the mitochondrial acyl carrier protein (mtACP). Holo-mtACP is formed in a post-translational modification reaction that consumes CoA as the apo-mtACP is transformed into its active holo form. Specifically, they show that low CoA levels caused by a defect in the first enzyme of CoA biosynthesis, pantothenate kinase (PanK), leads to low levels of holo-mtACP, which in turn causes decreased lipoylation of pyruvate dehydrogenase (PDH), and consequently lower PDH activity. These changes lead to specific phenotypes in the *Drosophila* model used in the study. The authors show that these phenotypes are largely reversed by either increasing CoA levels (through supplementation of a CoA biosynthetic intermediate) or by stimulating the activity of PDH. Similar effects were seen in human cell lines. Taken together, the authors propose a mechanistic model for the neurodegenerative disorders caused by defects in CoA/PDH pathways.

The study provides extensive data - especially from the *Drosophila* model - to support the specific conclusions that are drawn. In general, these conclusions seem valid and consistent with the data. However, the study does raise several questions that are not addressed in any of the experiments:

1. Lowered PDH activity as the unique correlator of CoA deprivation: As the authors indicate in their introduction, PDH is not the only mitochondrial protein that requires lipoylation for activity; others include the alpha-ketodehydrogenase active in the Krebs cycle. In addition, lowered holo-mtACP (the primary result of lowered CoA levels) will have several effects beyond just lipoylation; as mentioned in the discussion section, holo-ACP is also involved in iron-sulfur cluster biogenesis, among other functions. Indeed, the results of Jeong et al (the co-submitted manuscript) indicate that the reduction in CoA levels by a PanK defect also decreases the activity of aconitase and Complex I in their mouse model. Do the authors believe that their data convincingly excludes the possibility that defects in other enzymes/pathways are least contributing to the observed phenotypes in their model?

2. Understanding the effect of PDH stimulation when basal CoA levels are low: The authors provide experimental evidence from which they conclude that stimulation of PDH activity overcomes the deleterious effects of low CoA levels. They explain this by stating: "We assumed that a fraction of the remaining pool of lipoylated PDH in the affected wing is inactivated by the action of PDK or SIRT4 (a lipoamidase that also inhibits PDH) (Mathias et al., 2014) and therefore the remaining activity of PDH could be enhanced by interfering with the action of these inhibitory enzymes." However, stimulating PDH would not have any effect on the absolute levels of CoA (in all its form) and therefore the levels of holo-mtACP will remain low. Consequently, other pathways dependent on the active ACP will remain compromised. Instead, PDH activation would result in a modification of the CoA/AcCoA ratio in favour of the latter; however, the authors' mechanistic model does not

explain why the increased availability of AcCoA has the observed positive effects. As written, there is a convolution of effects associated with i) low CoA levels, and consequently low holo-mtACP levels, and ii) low PDH activity. The authors want to draw a direct correlation between the two, but the only experiment done in support of this resulted in quite a small effect (Fig. 2E). This experiment would have to be extended to gain further evidence of the direct link. As it stands, the authors should be more circumspect about how they interpret the results and draw conclusions in the context of this link actually not being as strong as they would want to suggest. Moreover, they should clarify their writing in regard to differentiating between effects directly linked to low CoA levels (such as low holo-mtACP) and those that follow from these (such as low PDH activity because of impaired lipoylation).

3. Correlation with human cell lines: The authors provide experiments in HEK293T and neuroblastoma cells as evidence that the effects they see in insect cells and in their model also occur in mammalian cells. While this seems to be true in regard to mtACP, it is not clear what the authors are implying with this result. That neuroblastoma cells are more sensitive to lower CoA levels than HEK293T cells? If so, this might be a result of the downregulation of PANK2 not being equally efficient in both cell lines, considering the data shown in Suppl. Figs 7 and 8. Also, the authors do not explain the identity of the second band clearly visible in their Western blot analysis of PanK2 expression in Fig. 6A & C. Considering the poor resolution of this band and the one of interest, can they be confident in the densitometry analysis shown in B & D? [While not mentioned elsewhere in this review, the same issue holds for the Western blot analysis shown in Fig. 2D, where multiple bands are also observed but not explained.] Are the authors implying that lower CoA levels will consequently also lead to lower PDH activity levels in these cells? They do not provide evidence of this. Consequently, the value of this specific experiment in the context of the study as the whole is not immediately clear.

4. Hypothesis of PKAN pathogenesis: As stated in the review of the manuscript by Jeong et al, the authors propose a model/hypothesis for PKAN pathogenesis based on their findings that fail to explain how the redundancy in pantothenate kinases rescues the defect in PanK2 in all organs except the brain. There are several possibilities, but unfortunately the current study cannot provide clarity in the regard, specifically as the model system used (*Drosophila*) is not ideal for answering such questions: it has a single pantothenate kinase, and therefore defects in this enzyme will have the observed global effects. It is important that the authors point out this shortcoming in the discussion.

While the comments above may sound quite negative, I found the manuscript by Lambrechts et al. to make a significant contribution to our understanding of PKAN and other similar neurodegenerative disorders. However, as pointed out above the authors should clearly explain how they delineate the various observed effects, and they should perhaps concede that their results do not give us the full picture as yet. Nonetheless, the possibility of using DCA for the treatment of PKAN is an important finding that could have far-reaching implications. However, the authors acknowledge that much work needs to be done to determine the clinical potential of such a treatment.

Overall, I would suggest that the manuscript be considered for publication once the matters above are fully addressed. I do not believe that co-publication with the manuscript by Jeong et al is necessary, or that failure to link the two publications will be detrimental to the understanding of either.

1st Revision - authors' response

22 August 2019

Comments by the editorial board: You will see from the comments pasted below that both referees enjoyed the article and have similar and overlapping concerns. The fly model remains a fly model in that it doesn't cover the mammalian complexity and by not investigating in other directions than the PDH enzyme, you are restricting the conclusiveness of the data and somehow weakening your study as compared to the accompanying article (see below). Further, the study needs to be considerably improved in terms of clarity, providing details and explanations and a better discussion along with high resolution WB.

We appreciate the reviewers' suggestions to address the point that the fly model is less complex compared to mammalian models. We have emphasized and clarified that the fly model allows

genetic and biochemical experiments in a living organism and that these tools allowed us to investigate our hypothesized CoA-mtACP-PDH link which explains the similarity of clinical symptoms of PKAN, CoPAN, MePAN and PDH-E2 deficiency. The findings by Jeong et al. are consistent with our findings, and their findings can be explained by one of the key findings of our manuscript: impaired CoA biosynthesis leads to decreased levels of holo-mtACP. We have rewritten the manuscript to better explain this in the text and improve its flow. We also have now included data showing additional lipoylation defects in addition to those in PDH (new Figure 3D, Appendix Figure S1). We have provided additional high-resolution western blots to validate the PANK2 downregulation in mammalian cells and to identify the specific PANK2 signal (Appendix Figure S10 including an explanatory figure legend). In addition, we received extensive feedback from non-Drosophila native English speakers to improve the overall clarity and readability. Finally, we have discussed in more detail the utility of Drosophila, its limitations, and how it differs specifically relevant to the genetics and diseases presented.

Following our cross-commenting exercise, ref 2 says: "explaining (if they can) why PDH deserves to be the singular focus of their studies, and potential future treatment are the key points to bring this ms closer to being acceptable for publication." This referee also notes that "attaching [their paper] to the Hayflick manuscript may actually reduce its impact, as it would highlight the limitations of the fly model (compared to a mammalian one) and also highlight the complexity of the defect's effects, specifically that it involves much more than just PDH."

We thank the reviewers for drawing our attention to the need to clarify our focus on PDH. We initially focused primarily on PDH because regulation of the activity of PDH is better understood than that of α -ketoglutarate dehydrogenase, the glycine cleavage system, or the branched chain amino acid pathways (Rowland et al., 2018). In addition, reliable Drosophila antibodies are available that detect PDH but not the other complexes. Moreover, the clinical symptoms of PDH-E2 deficiency are strikingly comparable to clinical symptoms of PKAN. Despite this, we agree that the other lipoylated enzymes may contribute and therefore we now have extended our focus to include other lipoylation defects. To clarify this, we provide explanations in the introduction and results section why we have focused here on PDH. We have also now included in the discussion suggestions for potential future treatments for specific diseases along the proposed CoA-mtACP-PDH axis; for PKAN we suggest 4-phosphopantetheine; for CoPAN we suggest DCA, SIRT4 inhibitors and an iron chelator; for MePAN and PDH-E2 deficiency we suggest DCA and SIRT4 inhibitors. These suggestions are based on our results, the results of Jeong et al. and the results of a recent manuscript (Klopstock et al., 2019). We believe that with these modifications and additions, the complexity of the defect is better highlighted, and the added value of jointly publishing the two manuscripts is better justified.

Therefore, I would like to invite you to revise your article and would encourage you to add more experimental evidence to strengthen the findings as much as you can. I believe that both articles strengthen one another, so would advise against dissociation. But therefore, further data are needed to bring your article to that level.

We have substantially revised our manuscript, added experimental evidence to further substantiate our findings, including rescue of PDH activity by DCA and CoA in Drosophila S2 cells, defects in lipoylation of other proteins, and additional Western blots for human PANK2 and CoA measurements.

Referee #1

The fly model system is well established, but the authors did not explain their interpretations well or describe the overall biological pathway clearly.

We have rewritten the manuscript, which has been substantially edited by a non-Drosophila native English speaker to improve the overall clarity. We also now have added biological pathways relevant for our study and relevant branches thereof (new Figure 1) as an illustration to provide foundational information for readers and better clarify our findings.

This paper was submitted at the same time as a paper by Jeong et al. It is doubtful that this paper could stand alone at the moment, because it is unclearly written, and much of the terminology is

directed to a fly audience, and the biochemical pathways are poorly described, and they completely ignored the potential role of 3 other enzyme systems in which lipoylation is required. It seems unlikely that their efforts to modify PDH could compensate, when alpha ketodehydrogenase, the glycine cleavage system, and the branched chain amino acid pathways are presumably also compromised.

We appreciate reviewer 1 drawing attention explicitly to these concerns, which we discuss below and have tried to address fully in our paper. To complement and explain the observations of Jeong et al., our work provides direct evidence for decreased 4'-phosphopantetheinylation of mtACP. Similarly, the clinical relevance of our observations is demonstrated by Jeong et al. Therefore, joint publication of the two manuscripts remains our preference.

We have rewritten the manuscript to better explain our points and to make it more accessible to all scientific audiences. In addition, we have added experiments demonstrating lipoylation defects in other proteins in addition to PDH (Figure 3D) and described the relevant biochemical pathways in more details. We have provided figures to illustrate the complexity and interconnection of the pathways. We thank the reviewer for emphasizing this point and directing us to undertake additional experiments. Though they have not entirely clarified the metabolic landscape in these disease states, they have broadened our focus and allowed for a more informed discussion.

We have added to the discussion that decreased activity of α KGDH, GCV system, and the branched chain amino acid pathways may also influence the observed phenotypes. SIRT4 regulates PDH activity via its lipoamidase activity, which cleaves the lipoyl moiety from the E2 component of PDH and thereby inhibits PDH activity (Mathias et al., 2014; Rowland et al., 2018) Therefore SIRT4 downregulation can increase PDH activity, explaining the rescue we observe. In addition, SIRT4 appears to interact also with α -ketoglutarate dehydrogenase (α KGDH) and branched-chain alpha-ketoacid dehydrogenase (BCKDH) (Mathias et al., 2014). Therefore, it is possible that SIRT4 can negatively influence lipoylation of these other enzymes as well, potentially explaining the potent rescue after SIRT4 downregulation. We now have added this to the discussion.

Specific comments ref# 1

1. the abstract is confusing and potentially misleading. What does the sentence "impaired CoA homeostasis leads to decreased 4 phosphopant... really mean? Their point in most experiments is that 4 phosphopant.. is required for CoA synthesis. the wording is circular. Also, by activation the "most downstream" is an unsupported assertion. What about the TCA cycle component and the other two active enzyme systems that require lipoylation? Aren't they more downstream? were they affected?

We have more carefully focused our attention on distinguishing between "4-phosphopantetheine" and "4'-phosphopantetheinylation". 4-phosphopantetheine is an intermediary product in the canonical coenzyme A biosynthesis route (Figure 1). Previously we demonstrated that when the pantothenate kinase step is blocked (as in PKAN), CoA can be formed from this intermediate (Srinivasan et al., 2015). Jeong et al. now show in their manuscript that 4'-phosphopantetheine also rescues PKAN disease phenotypes in their mouse model. In contrast, 4'-phosphopantetheinylation refers to the post-translation modification of proteins and for which coenzyme A is required, because the source for the 4-phosphopantetheine moiety is a coenzyme A molecule. Therefore, when a protein is 4-phosphopantetheinylated, coenzyme A is "consumed". Based on this, we hypothesized that cells may respond to impair de novo synthesis of CoA by inhibiting processes that consume CoA. Because 4'-phosphopantetheinylation consumes CoA, this process may be inhibited, leading to decreased 4'-phosphopantetheinylation of target proteins. Inhibiting a process that consumes CoA will result in less demand for CoA, potentially stabilizing levels. This will come with costs, namely decreased 4'-phosphopantetheinylation. Our hypothesis has not yet been tested however. Here we show that indeed decreased levels of human PANK2 and Drosophila PANK, which are required for de novo biosynthesis of CoA, do lead to decreased levels of 4'-phosphopantetheinylation of mtACP. We now have explained this more fully and have added extra schemes in Figure 1 to clarify these points. We have also deleted the wording "most downstream" and provided a more accurate description of the observations and predictions. In addition, we have demonstrated broader lipoylation defects in addition to that of

PDH and again indicate our appreciation for the attention to this by our reviewers.

They never explained the difference between de novo FA synthesis in mitochondria vs. FA consumption. A diagram and explanation with the full names of the enzymes and relevant intermediates would help here.

We now have explained this more fully in the revised Figure 1. In addition, we have extended the introduction, explaining in more detail the metabolic reactions in which CoA is recycled, including de novo fatty acid synthesis and degradation, occurring in mitochondria and the TCA cycle. Recognizing the complex interplay of all of these processes, we have placed more attention on clearly presenting the relevant pathways and processes.

2. They should have included a reference and explanation for the role of LIAS and included a strong reference, such as one by Squire Booker.

We thank the reviewer for this information and have now included the following references and explained the role of LIAS in the text.

-Booker SJ Chemistry and Biology 2004 (Booker, 2004)

-Hiltunen et al., Autio, Kastaniotis, BBA, 2010 (Hiltunen et al., 2010)

-Solomonson and DeBerardinis, Lipoic acid metabolism and mitochondrial redox regulation, JBC 2018. Thematic mini review (Solomonson & DeBerardinis, 2018)

3. In figure 2, they did not explain the relevance and meaning of COO enclosed in a red circle. In general, the figure legends needed a stronger narrative and better presentation.

We agree that the "COO" is confusing and have now omitted it from the figure and changed the red circle, clarifying that the holo-form and S99D and S99E are all negatively charged and that S99A is not. We have explained what the red circle indicates in the figure legend. All figure legends have now been modified to improve the clarity of the manuscript.

3. In a full section of the paper, they seem to think that PDH is the only enzyme that should be discussed and might contribute to the phenotype. They need to discuss this with more intellectual rigor.

As noted, we appreciate this suggestion and have now included data showing lipoylation defects in other enzyme complexes as well. We discuss a possible influence of the other lipoylated enzymes in the introduction, the results section, and the discussion.

6. In their fly kos, they need to convert to language that molecular medicine readers can easily follow. Instead of writing out UAS GFP etc. they need to point out which lines have a knockout of which gene in language accessible to most biologists. The full explanation can be included in the M&M,

We have adjusted the text to be accessible to the broader scientific community. We have received input from non-Drosophilists to modify this text. Full information of all the genetics and crosses is now provided in the Materials and Methods and in the Appendix file.

7. The notion that hypoxia would lead to the pathophysiology is not correctly represented. It would be that oxygen availability is normal, but respiration is impaired. The reviewer did not have time to review that paper, but confusing hypoxia with impaired respiration is a common mistake, and it needs to be clarified here.

We have removed this section from our discussion.

8. Should specify that activation of iron responsive factors was in yeast model system and name them.

We have adjusted this accordingly in the discussion.

9. Does the final sentence contradict the conclusions of the companion paper. No disease modifying treatments currently available seems inappropriate since the other paper described a strong

candidate.

In the discussion we conclude that 4-phosphopantetheine has therapeutic potential for PKAN, as was also proposed in Srinivasan et al 2015 (Srinivasan et al., 2015). Our suggestions for therapeutic possibilities for the other diseases are also now presented in the discussion of our revised manuscript.

10. Better description of figure 4 needed- what does the box demonstrate?

The box in the original Figure 4C is the area that is enlarged visible in 4D, which we now explain in the revised figure legend. For better general readability, we have moved some parts of the main figures to the Appendix files. The experiments performed in the larval imaginal wing discs mainly served as controls to demonstrate the effectiveness of the RNAi and the specificity of the antibodies and the specific drivers. We have explained this now more clearly in the Appendix file. With this we believe that the readability of the main text and understanding of the figures have improved.

11. Figure 6, the western of hPANK2 is not of acceptable quality. There is a huge apparently non-specific band. It is possible to do better mammalian westerns for PANK2.

We have added Western blots using an independently raised antibody against human PANK2 (gift from Prof. Jackowski). This antibody recognizes a band migrating at the same mobility as the band that is recognized by the commercially available antibody obtained from Origene and that corresponds to the band that is diminished after RNAi. Together we are able to identify the specific PANK2 band and demonstrate that the RNAi against human PANK2 was successful in downregulating human PANK2 in HEK293 cells and in the SH-SY-5Y cells. These Western blot results were consistent with our qPCR results. We have included these data in the revised manuscript. (Appendix Figure S10, including an explanatory figure legend). We note that commercially available antibodies from Sigma only recognize human PANK2 when it is overexpressed (see added blots Appendix Figure S10). The commercially available antibody from Origene detects the specific endogenous band representing human PANK2 but also non-specific bands.

12. The assertion that mtACP is done in the two PANK2 kd lines is not strong. Better data and more experimental points are needed here.

This is an important point raised by reviewer 1. We tried to generate data at multiple timepoints in our initial experiments but were limited in the results we obtained. These experiments will be repeated but more results will not be available in the timeframe needed for this manuscript. Therefore, we have softened our conclusions based on the data we have in hand, which show that downregulation of PANK2 leads to a decrease in mtACP in both neuroblastoma-derived cell lines and in one of the non-neuronal cell lines.

13. They should have explained the potential significance of hyperintensity in their discussion.

We have added a possible explanation to the introduction and discussion.

Overall, this paper is difficult to read, contains some weak data, and somehow ignores everything that would be affected other than PDH, which is not intellectually sufficiently rigorous for the anticipated reading audience. Some sentences were so confusing that this reviewer gave up. A strong rewrite would make a big difference.

Again, we appreciate the opportunity to substantially improve the presentation of this work which we think more clearly conveys its importance and strengthens its impact.

Referee #2 (comments)

The model is perfectly adequate for testing the questions that the researchers set to answer. However, it is limited in explaining how the results would translate to a mammalian system, as the relevant genetic defect is in a gene that encodes an enzyme (pantothenate kinase/PanK) that occurs as multiple isoforms in humans, of which the relevant PanK localizes to mitochondria. In contrast, *Drosophila* only has a single PanK. This makes for a simpler system that gives rise to the pronounced phenotypes that made the study possible. However, the lack of complexity also makes it impossible to predict how the results would translate to the mammalian system where subcellular differentiation is at play. See review for further comments in this regard.

This is an important point, and we note that, while *Drosophila* does have 1 PANK gene, it encodes various isoforms, one of which localizes to mitochondria (Wu et al., 2009). This indicates that at least part of the observed phenotype in *Drosophila* is caused by defects in PANK expression in the mitochondria. We have addressed this and explained this now more clearly in the revised manuscript. We then extend our results by corroborating them in a mammalian system in which we specifically downregulate PANK2 in different human cell lines. Our result shows a decrease in holo-mtACP as a result of PANK2 downregulation in neuroblastoma and HEK293 cells. Our observations provide direct evidence to explain the results obtained by Jeong et al. As discussed by Jeong et al. (and in their Figure 8), their observations can be explained by a decrease in 4'-phosphopantetheinylation and thereby a decrease of mtACP. While *Drosophila* is a less complex system, it is suitable to investigate the hypothesis we propose. We also now directly address the limitations of *Drosophila* regarding the 1 PANK gene versus the 4 PANK genes in humans.

Specific comments Referee # 2

1. Lowered PDH activity as the unique correlator of CoA deprivation: As the authors indicate in their introduction, PDH is not the only mitochondrial protein that requires lipoylation for activity; others include the alpha-ketodehydrogenase active in the Krebs cycle. In addition, lowered holo-mtACP (the primary result of lowered CoA levels) will have several effects beyond just lipoylation; as mentioned in the discussion section, holo-ACP is also involved in iron-sulfur cluster biogenesis, among other functions. Indeed, the results of Jeong et al (the co-submitted manuscript) indicate that the reduction in CoA levels by a PanK defect also decreases the activity of aconitase and Complex I in their mouse model. Do the authors believe that their data convincingly excludes the possibility that defects in other enzymes/pathways are least contributing to the observed phenotypes in their model?

The points raised by both reviewers have inspired us to modify our discussion as well as perform additional experiments in which we actually did measure CoA levels, as explained below.

Consequences of impaired coenzyme A de novo biosynthesis are complex and intermingled because coenzyme A is involved in numerous reactions. We show reduced PDH activity as one of the consequences of impaired CoA de novo biosynthesis, and we agree that this is not a unique correlate of altered CoA biosynthesis. We now show lipoylation defects in other enzyme complexes as well. The rescue by DCA and SIRT4 downregulation suggest that at least part of the downstream consequences of impaired CoA biosynthesis are associated with reduced PDH activity. However, it should be noted that SIRT4 interacts not only with PDH but also with alpha-ketoglutarate hydrogenase (α KGDH) and branched-chain alpha-keto acid dehydrogenase (BCKDH) (Mathias et al., 2014). Therefore, SIRT4 may negatively influence lipoylation of these other enzymes as well. This may explain the potent rescue after SIRT4 downregulation. We have added this to our discussion. For effects on aconitase and complex 1, which are expected to be also affected by decreased levels of holo-mtACP, see also our response to point 2.

While the Jeong et al. manuscript does show decreased activity of aconitase and complex I in a PANK2 defective background, they do not show an actual reduction in CoA levels but only that the synthesis enzymes are downregulated. Therefore, it cannot be concluded that PANK2 depletion causes a decrease in CoA levels. It may be that PANK2 affects levels of CoA only in specific subcellular compartments, as Jeong et al. suggest. It also may be that PANK2 depletion decreases the rate of CoA de novo biosynthesis and, in order to compensate for this, a cellular response is induced that lessens CoA consumption. Such a process is 4'-phosphopantetheinylation, and by reducing this consumptive process, total CoA levels might be stabilized. Decreased activity of aconitase and complex I as observed by Jeong et al. can be explained solely by a decrease in holo-mtACP, which can occur in the presence of normal total

CoA levels. We have now added additional experiments in which we measure levels of CoA in the human cell lines in which PANK2 is downregulated. These experiments demonstrate that levels of total CoA are not significantly decreased. We cannot exclude that levels of CoA in these cells are decreased at a subcellular level, such as in specific compartments of mitochondria. On one hand, these novel data add more complexity to the discussion; however, on the other, these novel data fit in our proposed model and are consistent with the findings of Jeong et al. This is described more below and is added to the discussion of the manuscript.

Our hypothesis predicts that a CoA consuming process, like 4'-phosphopantetheinylation, could be limited under circumstances of reduced CoA as well as under conditions of normal CoA levels. Reducing 4'-phosphopantetheinylation could be a strategy to keep total cellular CoA at a fixed level by decreasing CoA consumption. The phenotypes observed by Jeong et al. could therefore be caused by reduced holo-mtACP and not by an actual decrease in CoA levels. A reduction of 4'-phosphopantetheinylation will result in reduced levels of mtACP and as a consequence reduced iron-sulfur cluster formation, reduced aconitase activity and reduced Complex I activity. These observations are consistent with all results in both manuscripts. An explanation for the rescue by 4'-phosphopantetheine may be that its addition can rescue the CoA biosynthesis rate in the PANK2 defective background, recovering the CoA consuming 4'-phosphopantetheinylation process and normalizing levels of holo-mtACP. We now make a more explicit distinction between impaired CoA biosynthesis and CoA levels. We also explain that impaired CoA biosynthesis by PANK2 depletion does not necessarily have to lead to a decrease in total CoA levels. We have presented these additional data and discussed these new results in light of the results of the Jeong et al. manuscript.

2. Understanding the effect of PDH stimulation when basal CoA levels are low: The authors provide experimental evidence from which they conclude that stimulation of PDH activity overcomes the deleterious effects of low CoA levels. They explain this by stating: "We assumed that a fraction of the remaining pool of lipoylated PDH in the affected wing is inactivated by the action of PDK or SIRT4 (a lipoamidase that also inhibits PDH) (Mathias et al., 2014) and therefore the remaining activity of PDH could be enhanced by interfering with the action of these inhibitory enzymes." However, stimulating PDH would not have any effect on the absolute levels of CoA (in all its form) and therefore the levels of holo-mtACP will remain low. Consequently, other pathways dependent on the active ACP will remain compromised. Instead, PDH activation would result in a modification of the CoA/AcCoA ratio in favour of the latter; however, the authors' mechanistic model does not explain why the increased availability of AcCoA has the observed positive effects. As written, there is a convolution of effects associated with i) low CoA levels, and consequently low holo-mtACP levels, and ii) low PDH activity. The authors want to draw a direct correlation between the two, but the only experiment done in support of this resulted in quite a small effect (Fig. 2E). This experiment would have to be extended to gain further evidence of the direct link. As it stands, the authors should be more circumspect about how they interpret the results and draw conclusions in the context of this link actually not being as strong as they would want to suggest. Moreover, they should clarify their writing in regard to differentiating between effects directly linked to low CoA levels (such as low holo-mtACP) and those that follow from these (such as low PDH activity because of impaired lipoylation).

We agree with this reviewer that there is a convolution of effects associated with low CoA levels and possibly also with a reduction of CoA biosynthesis, even under circumstances when total CoA levels may not be affected. The full complexity of the effects can only be understood when it will be possible to measure subcellular level CoA levels, preferably using in situ methods.

We do agree with the reviewer that although this is an attractive model, explaining the results by us and Jeong, alternative explanations are possible and a linear consequence of impaired CoA biosynthesis on PDH activity needs to be demonstrated. This is now stated in the discussion along with a cautionary note in translating our results to the mammalian system.

We further agree with the reviewer that by stimulating PDH, other functions of holo-mtACP will remain low and may not be rescued. We have addressed this now in our discussion and the following paragraph was added:

"This rescue is somewhat surprising because boosting these enzyme activities still does not resolve decreased CoA levels or other effects of decreased levels of holo-mtACP. mtACP serves numerous

functions that are conserved across species. Indeed, NDUFAB1, the human ortholog of mtACP is a subunit of and required for the assembly of complex I (Vinothkumar et al., 2014; Van Vranken et al., 2018) and is involved in iron-sulfur biogenesis (Van Vranken et al., 2016). It is possible that the majority of the RNAi-induced phenotype in the Drosophila wing arise from reduced PDH activity and not from other affected processes downstream from holo-mtACP”.

We have performed additional experiments in S2 cells to further substantiate a link between impaired CoA biosynthesis and decreased activity of PDH. We now show that addition of DCA to the medium of S2 cells stimulated PDH activity and addition of CoA to the medium also rescues PDH activity”.

3. Correlation with human cell lines: The authors provide experiments in HEK293T and neuroblastoma cells as evidence that the effects they see in insect cells and in their model also occur in mammalian cells. While this seems to be true in regard to mtACP, it is not clear what the authors are implying with this result. That neuroblastoma cells are more sensitive to lower CoA levels than HEK293T cells? If so, this might be a result of the downregulation of PANK2 not being equally efficient in both cell lines, considering the data shown in Suppl. Figs 7 and 8. Also, the authors do not explain the identity of the second band clearly visible in their Western blot analysis of PanK2 expression in Fig. 6A & C. Considering the poor resolution of this band and the one of interest, can they be confident in the densitometry analysis shown in B & D? [While not mentioned elsewhere in this review, the same issue holds for the Western blot analysis shown in Fig. 2D, where multiple bands are also observed but not explained.] Are the authors implying that lower CoA levels will consequently also lead to lower PDH activity levels in these cells? They do not provide evidence of this. Consequently, the value of this specific experiment in the context of the study as the whole is not immediately clear.

See our answer to point 11 of reviewer 1. These results show that PANK2 is equally downregulated in all generated clones when PANK2-RNAi is induced. We have explained this better and indicated what background bands are and what the specific bands are.

For the densitometry assays, only the specific band, representing PANK2, indicated with the red arrow was measured.

For figure 3D (in the new version) the specific band is also indicated, the PDH-E2 band corresponds with the predicted Mw, and the signal of lipoylated PDH-E2, co-migrates at the same speed as PDH-E2. We have now explained this better in the text.

4. Hypothesis of PKAN pathogenesis: As stated in the review of the manuscript by Jeong et al, the authors propose a model/hypothesis for PKAN pathogenesis based on their findings that fail to explain how the redundancy in pantothenate kinases rescues the defect in PanK2 in all organs except the brain. There are several possibilities, but unfortunately the current study cannot provide clarity in the regard, specifically as the model system used (Drosophila) is not ideal for answering such questions: it has a single pantothenate kinase, and therefore defects in this enzyme will have the observed global effects. It is important that the authors point out this shortcoming in the discussion.

We are aware that our study does not fully explain why specific brain regions are affected in PKAN (or in CoPAN, MePAN and PDH-E2 deficiency) patients. Redundancy of other pantothenate kinases is certainly not the only possible answer because humans (and fruitflies) have only one gene coding for COASY (affected in CoPAN) and specific brain areas are affected in these patients as well. One possible explanation derived from our results and model could be that these regions are selectively vulnerable to loss of PDH activity. Our results and model also provide an explanation of why PKAN and CoPAN patients suffer from iron accumulation and MePAN and PDH-deficiency patients do not. We have explained this more clearly in the discussion.

We are aware that the Drosophila model is simple compared to complex mammalian systems and have discussed this now more explicitly in the revised text.

While the comments above may sound quite negative, I found the manuscript by Lambrechts et al. to make a significant contribution to our understanding of PKAN and other similar

neurodegenerative disorders. However, as pointed out above the authors should clearly explain how they delineate the various observed effects, and they should perhaps concede that their results do not give us the full picture as yet. Nonetheless, the possibility of using DCA for the treatment of PKAN is an important finding that could have far-reaching implications. However, the authors acknowledge that much work needs to be done to determine the clinical potential of such a treatment.

We thank the reviewers for their fair and scientifically sound comments which are constructive. By addressing them we believe the manuscript has improved substantially. We now discuss in more detail the various effects and alternative explanations but also we make more clear that we cannot explain all consequences of impaired CoA metabolism. Nevertheless, we believe that we have added important insights to help delineate the molecular pathogenesis of the disorders discussed. We also have included a brief discussion of rational therapeutics in all four disorders.

Overall, I would suggest that the manuscript be considered for publication once the matters above are fully addressed. I do not believe that co-publication with the manuscript by Jeong et al is necessary, or that failure to link the two publications will be detrimental to the understanding of either.

We believe that all matters have been addressed and we hope it will be decided that the two publications can remain linked.

References

- Booker, S. J. (2004). Unraveling the pathway of lipoic acid biosynthesis. *Chemistry & Biology*, 11(1), 10–2. <https://doi.org/10.1016/j.chembiol.2004.01.002>
- Hiltunen, J. K., Autio, K. J., Schonauer, M. S., Kursu, V. A. S., Dieckmann, C. L., & Kastaniotis, A. J. (2010). Mitochondrial fatty acid synthesis and respiration. *Biochimica et Biophysica Acta (BBA) - Bioenergetics*, 1797(6–7), 1195–1202. <https://doi.org/10.1016/j.bbabi.2010.03.006>
- Klopstock, T., Tricta, F., Neumayr, L., Karin, I., Zorzi, G., Fradette, C., et al. (2019). Safety and efficacy of deferiprone for pantothenate kinase-associated neurodegeneration: a randomised, double-blind, controlled trial and an open-label extension study. *The Lancet. Neurology*, 18(7), 631–642. [https://doi.org/10.1016/S1474-4422\(19\)30142-5](https://doi.org/10.1016/S1474-4422(19)30142-5)
- Mathias, R. A., Greco, T. M., Oberstein, A., Budayeva, H. G., Chakrabarti, R., Rowland, E. A., et al. (2014). Sirtuin 4 Is a Lipoamidase Regulating Pyruvate Dehydrogenase Complex Activity. *Cell*, 159(7), 1615–1625. <https://doi.org/10.1016/j.cell.2014.11.046>
- Rowland, E. A., Snowden, C. K., & Cristea, I. M. (2018). Protein lipoylation: an evolutionarily conserved metabolic regulator of health and disease. *Current Opinion in Chemical Biology*, 42, 76–85. <https://doi.org/10.1016/j.cbpa.2017.11.003>
- Solomonson, A., & DeBerardinis, R. J. (2018). Lipoic acid metabolism and mitochondrial redox regulation. *Journal of Biological Chemistry*, 293(20), 7522–7530. <https://doi.org/10.1074/jbc.TM117.000259>
- Srinivasan, B., Baratashvili, M., van der Zwaag, M., Kanon, B., Colombelli, C., Lambrechts, R. A., et al. (2015). Extracellular 4'-phosphopantetheine is a source for intracellular coenzyme A synthesis. *Nature Chemical Biology*, 11(10), 784–92. <https://doi.org/10.1038/nchembio.1906>
- Vinothkumar, K. R., Zhu, J., & Hirst, J. (2014). Architecture of mammalian respiratory complex I. *Nature*, 515(7525), 80–84. <https://doi.org/10.1038/nature13686>
- Van Vranken, J. G., Jeong, M.-Y., Wei, P., Chen, Y.-C., Gygi, S. P., Winge, D. R., & Rutter, J. (2016). The mitochondrial acyl carrier protein (ACP) coordinates mitochondrial fatty acid synthesis with iron sulfur cluster biogenesis. *ELife*, 5. <https://doi.org/10.7554/eLife.17828>
- Van Vranken, J. G., Nowinski, S. M., Clowers, K. J., Jeong, M.-Y., Ouyang, Y., Berg, J. A., et al. (2018). ACP Acylation Is an Acetyl-CoA-Dependent Modification Required for Electron Transport Chain Assembly. *Molecular Cell*, 71(4), 567–580.e4. <https://doi.org/10.1016/J.MOLCEL.2018.06.039>
- Wu, Z., Li, C., Lv, S., & Zhou, B. (2009). Pantothenate kinase-associated neurodegeneration: insights from a Drosophila model. *Human Molecular Genetics*, 18(19), 3659–3672. <https://doi.org/10.1093/hmg/ddp314>

Thank you for the submission of your revised manuscript to EMBO Molecular Medicine. We have now received the enclosed reports from the referees that were asked to re-assess it. As you will see the reviewers are now globally supportive and I am pleased to inform you that we will be able to accept your manuscript pending the following final amendments:

1) Please address the minor changes commented by referees 1 and 2.

I look forward to reading a new revised version of your manuscript as soon as possible.

***** Reviewer's comments *****

Referee #1 (Remarks for Author):

The revised version of the paper is greatly improved. It represents an important contribution and contains many conceptual and pragmatic implications.

A major issue

1. Figure 7A does not show reduced mtACP in the gel. The statistics in 7B presumably include this experiment. Either there was a mistake in placement of the gel, or this figure and its statistics should be removed.

Minor

Figure 6 Strain misspelled in 6D

Klopstock ref does not appear in ref list.

Referee #2 (Comments on Novelty/Model System for Author):

Highlighting the limitations of the model system was one of the points that the authors had to address in revision; I am satisfied that they have now done this.

Referee #2 (Remarks for Author):

I appreciate the extent to which the authors tried to address the comments raised by the reviewers. While the revised manuscript is still lacking in some respects, this is more due to the complexity of the system than to any particular shortcoming on the side of the authors. I therefore support publication after the authors have addressed the following minor points:

1. Abstract: the line "We demonstrate that CoA-dependent activation of mitochondrial acyl-carrier protein (mtACP) is the common molecule linking these diseases through its effect on PDH activity." should be reconsidered. As phrased, they are referring to a process, not a molecule. Also, I question their assertion that they "demonstrate" the link; rather, they provide evidence that is consistent with such a link.

2. Introduction: Is it necessary to underline hypo and hyper? Surely the reader can make this distinction themselves. I found this more confusing than helpful.

3. Introduction, top of p. 3. Please rephrase the sentence: "PDH catalyzes the oxidative decarboxylation of pyruvate to produce acetyl and then connects it to CoA and hereby links glycolysis to the TCA cycle (Figure 1)." to "...oxidative decarboxylation of pyruvate to produce acetyl-CoA, thereby linking glycolysis...". Point is that "acetyl" is not a molecule; acetate is but referring to it in this context would be confusing.

4. Introduction, top of p. 3. Please reconsider the following sentence: "PKAN and Leigh disease are each in the differential diagnosis of the other..., for reasons of their phenotypic overlap." The meaning of this is unclear, specifically the phrase "disease are each in the diagnosis of the other". Do they mean there is a convolution or confusion of the two?

5. Introduction, p. 3" The phrase "Because CoA is recycled, these reactions do not lead to reduced CoA levels." should be clarified. Suggestion: "Because CoA is reused as acyl-carrying component, these reactions do not lead to reduced levels of total CoA".

6. Results, p. 5 and 6 (PDH-E2 lipoylation). The authors should be clearer that the new results focused on effects on total lipoylation (i.e. not on any specific protein) in the first instance. The way this section is currently written, the reader might assume that the lipoylation was targeted to the four mentioned proteins. This is somewhat clarified by the indication that the PDH-E2 antibody was used to check its lipoylation specifically, but it would be better to make this explicit at the outset. Also, the last sentence of this section could be more qualified by rather stating "..., but the observation that total protein lipoylation is reduced under conditions of impaired CoA biosynthesis, suggests that the other lipoylated fly enzymes are similarly affected."

7. CoA levels vs rate of biosynthesis - please be careful in distinguishing between these two phenomena. Using a phrase such as "decreased levels of CoA biosynthesis" (p.6) unnecessarily confounds them, and makes it unclear what is being referred to.

8. Fig. 1 and elsewhere: I understand that the authors use the term "recycling of CoA" to contrast it to the consumption of CoA in the process of holo-ACP synthesis. However, the nonspecialist might erroneously understand "recycling" to mean that it is broken down, and then salvaged. As indicated in comment 5 above, consider referring to the "reuse of CoA as acyl-carrying component" instead.

9. Reporting relative vs absolute data: while the use of relative data is warranted in some cases (eg. Fig. 3D, 7B, 7D) there is no need for this in others. In fig. 3F, the actual measured PDH activity should be reported, and in Fig. 7E and 7F the actual CoA concentrations. Using relative measures in these instances first hides useful information, and second unnecessarily complicates the calculation of the error as it requires the propagation of the error in the calculation of the ratio.

2nd Revision - authors' response

11 October 2019

Referee #1:

The revised version of the paper is greatly improved. It represents an important contribution and contains many conceptual and pragmatic implications.

A major issue

1. Figure 7A does not show reduced mtACP in the gel. The statistics in 7B presumably include this experiment. Either there was a mistake in placement of the gel, or this figure and its statistics should be removed.

The statistics indeed include this experiment, as well as 3 other independent experiments. We now have added a more representative blot, out of the 4 independent experiments reflecting better the conclusion based on the statistics, which remain as they are.

Minor Figure 6 Strain misspelled in 6D Klopstock ref does not appear in ref list.

We have changed this, and we have added Klopstock et al. to the reference list.

Referee #2:

Highlighting the limitations of the model system was one of the points that the authors had to address in revision; I am satisfied that they have now done this. Referee #2 (Remarks for Author): I appreciate the extent to which the authors tried to address the comments raised by the reviewers. While the revised manuscript is still lacking in some respects, this is more due to the complexity of the system than to any particular shortcoming on the side of the authors. I therefore support publication after the authors have addressed the following minor points: 1. Abstract: the line "We demonstrate that CoA-dependent activation of mitochondrial acyl-carrier protein (mtACP) is the common molecule linking these diseases through its effect on PDH activity." should be reconsidered. As phrased, they are referring to a process, not a molecule. Also, I question their

assertion that they "demonstrate" the link; rather, they provide evidence that is consistent with such a link.

We have changed this sentence into: "We provide evidence that CoA-dependent activation of mitochondrial acyl-carrier-protein (mtACP) is a possible process linking these diseases through its effect on PDH activity".

2. Introduction: Is it necessary to underline hypo and hyper? Surely the reader can make this distinction themselves. I found this more confusing than helpful.

We have removed the underlining of these words.

3. Introduction, top of p. 3. Please rephrase the sentence: "PDH catalyzes the oxidative decarboxylation of pyruvate to produce acetyl and then connects it to CoA and hereby links glycolysis to the TCA cycle (Figure 1)." to "...oxidative decarboxylation of pyruvate to produce acetyl-CoA, thereby linking glycolysis...". Point is that "acetyl" is not a molecule; acetate is but referring to it in this context would be confusing.

We have changed the text accordingly into: "PDH catalyzes the oxidative decarboxylation of pyruvate to produce acetyl-CoA, thereby linking glycolysis to the TCA cycle (Figure 1)".

4. Introduction, top of p. 3. Please reconsider the following sentence: "PKAN and Leigh disease are each in the differential diagnosis of the other..., for reasons of their phenotypic overlap." The meaning of this is unclear, specifically the phrase "disease are each in the diagnosis of the other". Do they mean there is a convolution or confusion of the two?

For more clarity, the sentences have been changed into: "The symptoms, signs and MRI characteristics of PKAN and PDH-E2 deficiency can be similar. In patients with a clinical suspicion of PKAN, PDH-E2 deficiency should also be considered in the differential diagnosis and vice versa. (Head et al., 2005; Leoni et al., 2012; McWilliam et al., 2010)".

5. Introduction, p. 3" The phrase "Because CoA is recycled, these reactions do not lead to reduced CoA levels." should be clarified. Suggestion: "Because CoA is reused as acyl-carrying component, these reactions do not lead to reduced levels of total CoA".

We have changed the text accordingly.

6. Results, p. 5 and 6 (PDH-E2 lipoylation). The authors should be clearer that the new results focused on effects on total lipoylation (i.e. not on any specific protein) in the first instance. The way this section is currently written, the reader might assume that the lipoylation was targeted to the four mentioned proteins. This is somewhat clarified by the indication that the PDH-E2 antibody was used to check its lipoylation specifically, but it would be better to make this explicit at the outset.

***We now have changed the heading of this section into: "Protein lipoylation is reduced by impeding CoA biosynthesis
And we have changed:
"To assess whether lipoylation was affected under conditions of impaired CoA biosynthesis, lipoylation was analysed using Western blot analysis. Incubation with an antibody against lipoylated proteins revealed decreased levels of various protein bands under conditions of reduced CoA levels, an effect that was rescued when CoA was supplemented to the medium (Figure 3D, Appendix Fig S1)."***

Into:

"To assess whether lipoylation was affected under conditions of impaired CoA biosynthesis, total protein lipoylation was analysed using Western blot analysis. Incubation with an antibody that detects protein-bound lipoic acid revealed decreased levels of various protein bands under conditions of reduced CoA levels, an effect that was rescued when CoA was supplemented to the medium (Figure 3D, Appendix Fig S1)."

Also, the last sentence of this section could be more qualified by rather stating "..., but the observation that total protein lipoylation is reduced under conditions of impaired CoA biosynthesis,

suggests that the other lipoylated fly enzymes are similarly affected."

We have changed the last sentence into: "Specific antibodies for the other lipoylated fly enzymes are lacking, but the observation that total protein lipoylation is reduced under conditions of impaired CoA biosynthesis, suggests that the other lipoylated fly enzymes are similarly affected".

7. CoA levels vs rate of biosynthesis - please be careful in distinguishing between these two phenomena. Using a phrase such as "decreased levels of CoA biosynthesis" (p.6) unnecessarily confounds them, and makes it unclear what is being referred to.

We have this now made clear throughout the text, indicated by track changes.

8. Fig. 1 and elsewhere: I understand that the authors use the term "recycling of CoA" to contrast it to the consumption of CoA in the process of holo-ACP synthesis. However, the nonspecialist might erroneously understand "recycling" to mean that it is broken down, and then salvaged. As indicated in comment 5 above, consider referring to the "reuse of CoA as acyl-carrying component" instead.

We have now replaced recycled and recycling by re-used or re-usage.

9. Reporting relative vs absolute data: while the use of relative data is warranted in some cases (eg. Fig. 3D, 7B, 7D) there is no need for this in others. In fig. 3F, the actual measured PDH activity should be reported, and in Fig. 7E and 7F the actual CoA concentrations. Using relative measures in these instances first hides useful information, and second unnecessarily complicates the calculation of the error as it requires the propagation of the error in the calculation of the ratio.

We agree with the reviewer that using relative measures can be meaningful in certain, but complicate matters in others. Regarding the PDH activity measurements in Figure 3F, as stated by the manufacturer of the Pyruvate Dehydrogenase (PDH) Activity Colorimetric Assay Kit (BioVision), multiple factors, like incubation time, room temperature, handling etc. influence the signals. As long as the samples and standard curve are measured under the same conditions, and the samples fall within the linear range of the standard curve, their measurements can be compared. Despite a similar trend in our experiments, we noticed considerable variability between separate experiments (N=5), likely due to the above described factors. In order to account for this interexperimental variability, we therefore normalized the samples to each control sample, measured within the same experiment. We therefore consider the normalized data in fig 3F to be the correct way to present the PDH activity measurements.

Regarding the CoA measurements in figure 7E and F, we have now added the actual CoA concentrations belonging to Figure 7E and F to Appendix Figure S10 A and B respectively.

Corresponding Author Name: Ody C.M. Sibon

Manuscript Number: EMM-2019-10488